# Electrically driven organic laser using integrated OLED pumping

Kou Yoshida[1,4], Junyi Gong[1,4], Alexander L. Kanibolotsky[2,3], Peter J. Skabara[2], Graham A. Turnbull[1✉] & Ifor D. W. Samuel[1✉]

Organic semiconductors are carbon-based materials that combine optoelectronic properties with simple fabrication and the scope for tuning by changing their chemical structure[1–3]. They have been successfully used to make organic light-emitting diodes[2,4,5] (OLEDs, now widely found in mobile phone displays and televisions), solar cells[1], transistors[6] and sensors[7]. However, making electrically driven organic semiconductor lasers is very challenging[8,9]. It is difficult because organic semiconductors typically support only low current densities, suffer substantial absorption from injected charges and triplets, and have additional losses due to contacts[10,11]. In short, injecting charges into the gain medium leads to intolerable losses. Here we take an alternative approach in which charge injection and lasing are spatially separated, thereby greatly reducing losses. We achieve this by developing an integrated device structure that efficiently couples an OLED, with exceptionally high internal-light generation, with a polymer distributed feedback laser. Under the electrical driving of the integrated structure, we observe a threshold in light output versus drive current, with a narrow emission spectrum and the formation of a beam above the threshold. These observations confirm lasing. Our results provide an organic electronic device that has not been previously demonstrated, and show that indirect electrical pumping by an OLED is a very effective way of realizing an electrically driven organic semiconductor laser. This provides an approach to visible lasers that could see applications in spectroscopy, metrology and sensing.

Organic semiconductors consist of conjugated molecules that can be simply deposited by evaporation or from solution to make a range of electronic and optoelectronic devices[1–7,12–15]. The properties of these semiconductors can be tuned by changing the chemical structure[16,17], and they are compatible with a wide range of substrates[18]. Their visible band gaps have made these semiconductors particularly suitable for applications in displays, and they are used across the world in mobile phones. Owing to these properties, they are used as laser materials. Furthermore, they have very high gain for lasing because of their strong optical transitions[11], and are widely tunable because of their broad spectra[19]. So far nearly all work on organic semiconductor lasers has involved optical pumping by another laser[9,10]. Although this has enabled some of the advantages above to be demonstrated, optical pumping is relatively complicated and expensive, and so electrical pumping of organic semiconductor lasers is desirable to enable their widespread use.

Electrical pumping involves injecting a current to generate a population inversion and hence light amplification, but this is difficult to achieve in organic semiconductors[10,11,20,21]. The low charge-carrier mobility of the materials[20,22,23] makes it difficult to inject the very high (kA cm⁻²) current densities needed, and causes an accumulation of injected charges (polarons) that have some absorption at the laser wavelength. The low mobility also makes it necessary to locate the

contacts very close to the gain medium, which can lead to additional loss. Furthermore, many injected charges will form triplets that do not contribute to light emission in fluorescent materials, but may absorb the light emitted and quench singlets. A previous study showed that the absorption spectrum of polarons and triplets of a carbazole-based laser material did not overlap with the gain spectrum[24], so one of the above problems could be overcome[25]. A device using this material showed some features of lasing, including the narrowing of the emission spectrum[25]. However, the emitted beam was not very clear, and the yield of these devices was low (5%) and their stability was very poor (operated for 20 pulses above the threshold).

Because of the difficulties outlined above, we have used an alternative approach in which we separate the region where the charges are injected from the region where the laser population inversion is formed. The gain medium is then excited by electroluminescence from the charge-injection region[26–28]. This is conceptually similar to diode-laser-pumped solid-state lasers and to nitride light-emitting-diode pumping[26,27], but here we achieve a fully integrated all-organic device. In this way, we avoid the losses due to injected charges, greatly reduce the losses due to triplets and also reduce the losses due to contacts. However, this approach has its own challenges such as the need for an organic light-emitting diode (OLED) with exceptionally high-intensity light output and the need to transfer the

[1]Organic Semiconductor Centre, SUPA, School of Physics and Astronomy, University of St Andrews, St Andrews, UK. [2]WestCHEM, School of Chemistry, University of Glasgow, Glasgow, UK. [3]Institute of Physical-Organic Chemistry and Coal Chemistry, Kyiv, Ukraine. [4]These authors contributed equally: Kou Yoshida, Junyi Gong. ✉e-mail: gat@st-andrews.ac.uk; idws@st-andrews.ac.uk

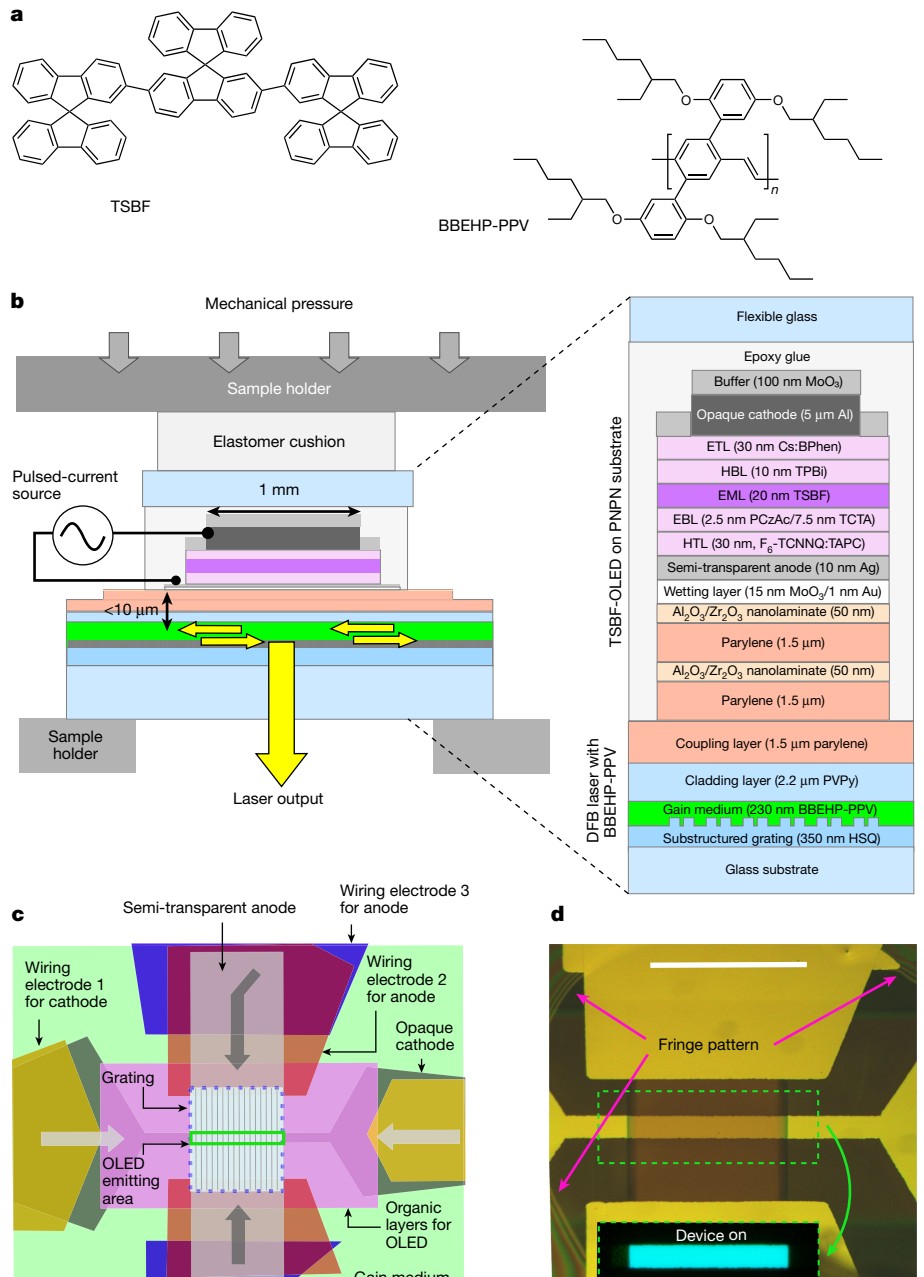

**Fig. 1 | Structure of the electrically driven organic semiconductor laser.**
**a**, Chemical structure of TSBF and BBEHP-PPV. **b**, Cross-section schematic of
the electrically driven laser; yellow arrows indicate laser feedback and output.
**c**, Schematic top view; grey arrows indicate lateral paths of injected charges.
**d**, Microscope image of the electrically driven laser. Inset, image of the laser
surface under operation. Scale bars: 1 mm (**c**,**d**). ETL, electron-transport
layer; BPhen, 4,7-diphenyl-1,10-phenanthroline; TPBi, 1,3,5-tris(1-phenyl-1H-

benzimidazol-2-yl)benzene; HBL, hole-blocking layer; EML, emission layer;
EBL, electron-blocking layer; PCzAc, 9,10-dihydro-9,9-dimethyl-10-(9-phenyl-
9H-carbazol-3-yl)-acridine; TCTA, 4,4′,4″-tris(carbazol-9-yl)triphenylamine;
HTL, hole-transport layer; $F_6$-TCNNQ, 2,2′-(perfluoronaphthalene-2,6-diylidene)
dimalononitrile; TAPC, 1,1-bis(4-(N,N-di(p-tolyl)amino)phenyl)cyclohexane;
PVPy, poly(vinyl-pyrrolidone); HSQ, hydrogen silsesquioxane.

electroluminescence generated into a high-density population inversion in the laser region. The former requires an improvement in OLED performance as well as the operation of the OLED in short light pulses. The latter requires a structure in which the OLED and laser cavity are very closely integrated, without impairing the performance of either.

## Overview of the integrated OLED

The integrated device (Fig. 1) consists of a multilayer stack of an OLED electroluminescence region, a central transparent light coupling region and a polymer distributed feedback (DFB) laser cavity. The structure,

therefore, has two light-emitting layers: an electroluminescent layer based on 2,7-bis(9,9-spirobifluoren-2-yl)-9,9-spirobifluorene (TSBF) and a stimulated emission layer based on poly(2,5-bis(2′,5′-bis(2″-ethylhexyloxy)phenyl)-p-phenylenevinylene) (BBEHP-PPV) (see Fig. 1a for the chemical structure of BBEHP-PPV and TSBF). BBEHP-PPV was selected as the laser gain medium because DFB lasers based on this polymer have shown some of the lowest lasing thresholds reported for organic lasers[7,8,29]. The absorption of BBEHP-PPV peaks around 430 nm, matching the emission spectrum of TSBF (Fig. 2a), so electroluminescence from TSBF can efficiently excite a population inversion in BBEHP-PPV.

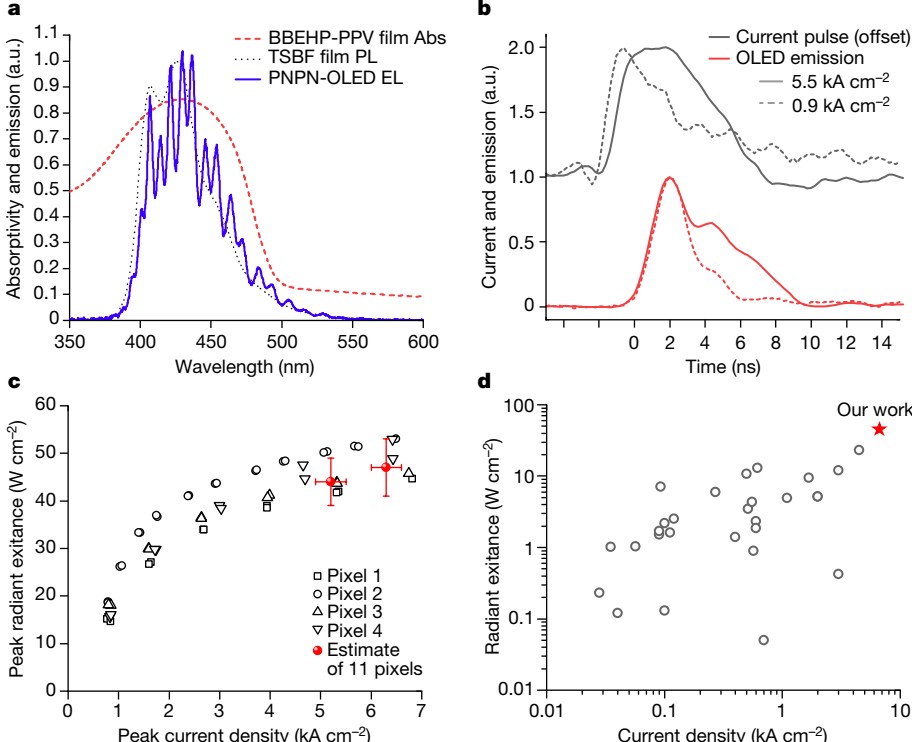

**Fig. 2 | Performance of TSBF-OLED on PNPN substrate. a**, Electroluminescence (EL) spectra of the OLED and comparison with the photoluminescence (PL) from a thin TSBF film and the absorptivity (Abs.) spectrum of a 230-nm thick BBEHP-PPV film. **b**, Time profile of driving current and electroluminescence of the OLED at different peak current densities. **c**, Peak radiant exitance of the OLED as a function of peak current density. Black symbols show the data of four different pixels fabricated in the same batch. Red symbols show the estimates of peak radiant exitance of the PNPN-OLED at 5.2 kA cm$^{-2}$ and 6.3 kA cm$^{-2}$, averaged over interpolated data of 11 pixels, including the four pixels shown in the figure and seven pixels from a different batch. Error bars are extended uncertainties of the measurement with a coverage factor of 2. **d**, Summary of radiant exitance of reported OLEDs and comparison with the OLED. a.u., arbitrary unit.

Emission from an OLED is usually highly divergent, so the irradiance decreases rapidly with distance. This is severe even over a short distance for OLEDs with a small active area. For the OLED of active area 130 µm × 1 mm used in this work, at a distance of 100 µm, we find that the peak irradiance is reduced to only 60% of the peak radiant exitance of the OLED, whereas at 7 µm, the peak irradiance is higher than 99% (Extended Data Fig. 1). We therefore designed the OLED and the laser waveguide to be separated by a distance of only 7 µm to maximize the excitation density in the gain material.

The organic laser and OLED sections were initially fabricated separately before being integrated to form the complete structure (Fig. 1b,c). The bottom-emitting OLED was deposited on a separate glass carrier that was first coated with two pairs of 1.5-µm-thick parylene-C layer (P) and 50-nm-thick Al$_2$O$_3$/ZrO$_2$ nanolaminates (N)[30] (Fig. 1b). The OLED and its PNPN substrate were then removed from the glass carrier (PNPN-OLED) for transfer onto the organic laser waveguide. The two pairs of P and N layers were used to give a better barrier to oxygen and moisture than would be provided by a single pair[30]. The laser consisted of a 230-nm thick BBEHP-PPV layer deposited on a distributed feedback grating; a 2.2-µm cladding layer of poly(vinyl-pyrrolidone) (PVPy) and a coupling layer of 1.5-µm parylene were coated on the BBEHP-PPV to complete the laser section. The two sections were held together mechanically using an elastomer cushion (MD700, Solvay) to ensure an even pressure and to make conformal optical contact between the two outer parylene layers. The fringe pattern seen under the microscope (Fig. 1d) indicates good conformal contact of the OLED and laser waveguide sections without any air gap in between.

A crucial aspect of the integrated design is that the transfer efficiency of electroluminescence to the laser gain medium can be much higher than the conventional outcoupling from an OLED—that is, emitting to air. This is because the similar refractive indices of the polymer layers between the OLED and the DFB laser ensure that minimal light is lost to substrate modes by total internal reflection, in contrast to the high (>70%) losses at an air interface. We simulated the outcoupling efficiency of the TSBF-OLED on the PNPN substrate emitting to media with different refractive indices (Extended Data Fig. 2) and found an outcoupling efficiency of the OLED to the PVPy layer of 62%, which is 2.3 times higher than the 27% outcoupling efficiency into the air. Thus our integrated device enables electroluminescence to be transferred very efficiently to the laser gain medium.

## OLED design for high light output

The threshold of our BBEHP-PPV lasers was around 100 W cm$^{-2}$, so even after taking into account the efficient coupling in an integrated structure, we need an OLED that would emit 50 W cm$^{-2}$ into the air—higher than what has been achieved until now. In previously reported OLEDs, radiant exitance around 20 W cm$^{-2}$ has been achieved at 4.5 kA cm$^{-2}$ with an external quantum efficiency (EQE) of 0.2% (ref. 31). For this efficiency, a very high current density, more than 10 kA cm$^{-2}$, would be needed to give a light output of 50 W cm$^{-2}$. To add to the difficulty of injecting such a high current density, we also need emission in the blue light region. For our desired emission wavelength of 430 nm, the highest radiant exitance reported is 4.3 W cm$^{-2}$ (ref. 32). The key design features that have enabled us to achieve record radiant exitance were to use an emitter with a short radiative lifetime, use a contact design that minimizes resistance, use intense short electrical pulses and use doped transport layers to facilitate charge injection and transport

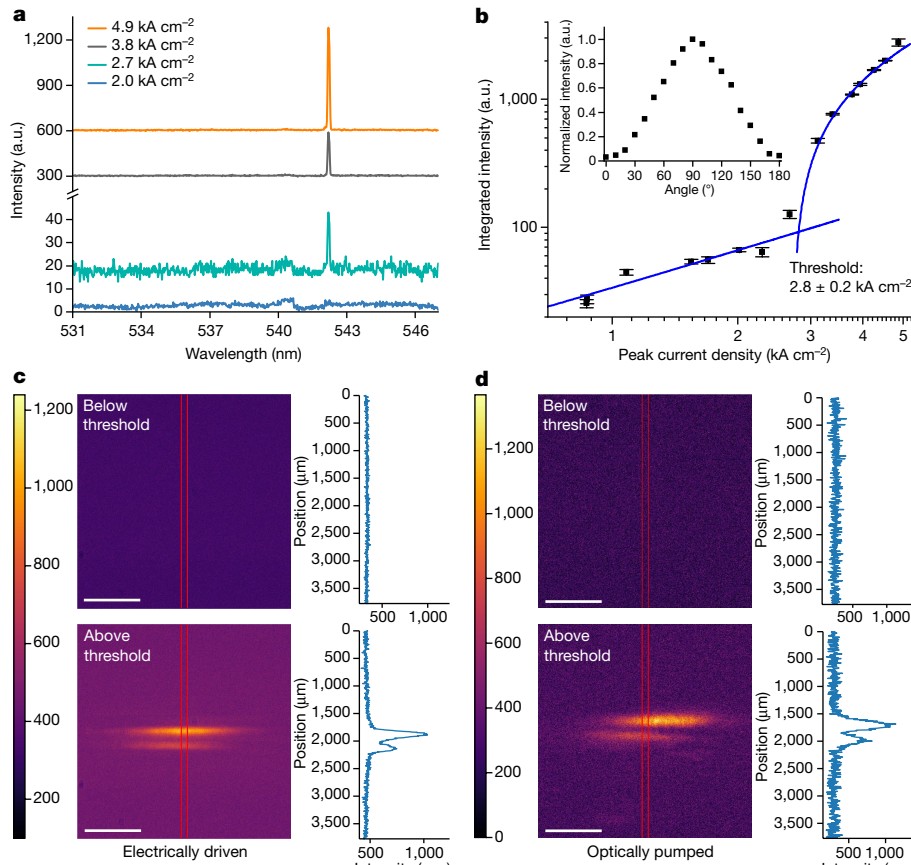

**Fig. 3 | Characterization of the integrated laser under electrically driven operation. a**, Evolution of the emission spectrum under different peak current densities below and above the threshold (offset for clarity). **b**, Integrated lasing intensity as a function of peak current density. Error bars show the standard error of the measurement. The blue lines are linear fits to the data below and above the threshold. Inset, normalized intensity as a function of the linear polarizer angle when the sample is electrically driven above the threshold; 90° is defined as the polarization parallel to the grating groove direction. **c**, Far-field emission image and averaged line-beam profile of electrically driven laser, measured below and above the threshold at 6 cm distance from the device. **d**, Corresponding far-field emission image and averaged line-beam profile of optically pumped laser. a.u., arbitrary unit. Scale bar, 1,000 μm (**c**,**d**).

(Methods). The high-intensity OLEDs have generally been very small, but this is not good for exciting a DFB laser because small excitation spots reduce the interaction length with the gain medium and increase the threshold[33,34]. Here we designed the OLED to be a narrow stripe with a length of 1 mm, to match the size of the laser grating, and a width of 130 μm, to have a small emitting area (but not so small that the divergence of the OLED would greatly reduce its intensity). This shape keeps the capacitance low and ensures that the current needs to travel only a very small distance across the semi-transparent contact, thereby reducing resistance, drive voltage and heating.

## OLED performance under pulse operation

The temporal profile of the optical pulse used to excite an organic laser is important because it affects the dynamics of the optical gain and hence the lasing threshold[33]. Figure 2b shows the time profiles of the driving current pulse and electroluminescence at different peak current densities. The shape of the current pulse was triangular at lower current densities, but became rectangular with a full width at half maximum of 5.4 ns at a peak current density of 5.5 kA cm$^{-2}$. The OLED could be operated (in pulsed mode) at a peak current density of 5.5 kA cm$^{-2}$ without breakdown even when the device was deposited on parylene-based substrates with poor thermal conductivity. At this current density, the rise time (10–90%) of the OLED output pulse was 1.5 ns, and the fall time was around 5.6 ns. The short rise time—only double the emission lifetime of BBEHP-PPV (0.72 ns)[35]—helps in achieving very high excitation

density in the laser. We note that the rise time obtained is much shorter than the 6-ns rise time of the larger area (approximately 1 mm$^2$) GaN light-emitting diodes used in our previous work[26,33,36].

Figure 2a shows the electroluminescence spectrum of the OLED under pulsed operation at a peak current density of 5.3 kA cm$^{-2}$ at 100 Hz. The spectrum has a maximum around 430 nm with a shape similar to the photoluminescence of TSBF, but modified by thin-film interference in the micrometre-thick PNPN substrate[30,37]. The electroluminescence spectrum gives a good overlap with the absorption spectrum of BBEHP-PPV, and we calculate that 75% of the light will be absorbed by a 230-nm thick BBEHP-PPV film.

Figure 2c shows the peak radiant exitance of the OLED as a function of peak current density. We observe an exceptionally high radiant exitance of 47 W cm$^{-2}$ at 6.3 kA cm$^{-2}$, which corresponds to a nominal EQE of 0.26%. We note that this value is underestimated because the measured peak current density includes a capacitive current component that does not contribute to charge recombination. Moreover, this is comparable to other OLEDs operated at such high current density (Extended Data Fig. 3). For example, at 100 A cm$^{-2}$, the reported EQE of an OLED is typically less than 1% and close to 0.1% for current density more than 1 kA cm$^{-2}$ (Supplementary Table 1). To our knowledge, our OLED has the highest intensity light output reported so far (Fig. 2d) and is also notable for achieving this at a very short wavelength of 430 nm (Extended Data Fig. 3f). A summary of previous work is available in the Supplementary Information. Furthermore, the very short output pulses reduce the problem of triplet generation in organic lasers[33].

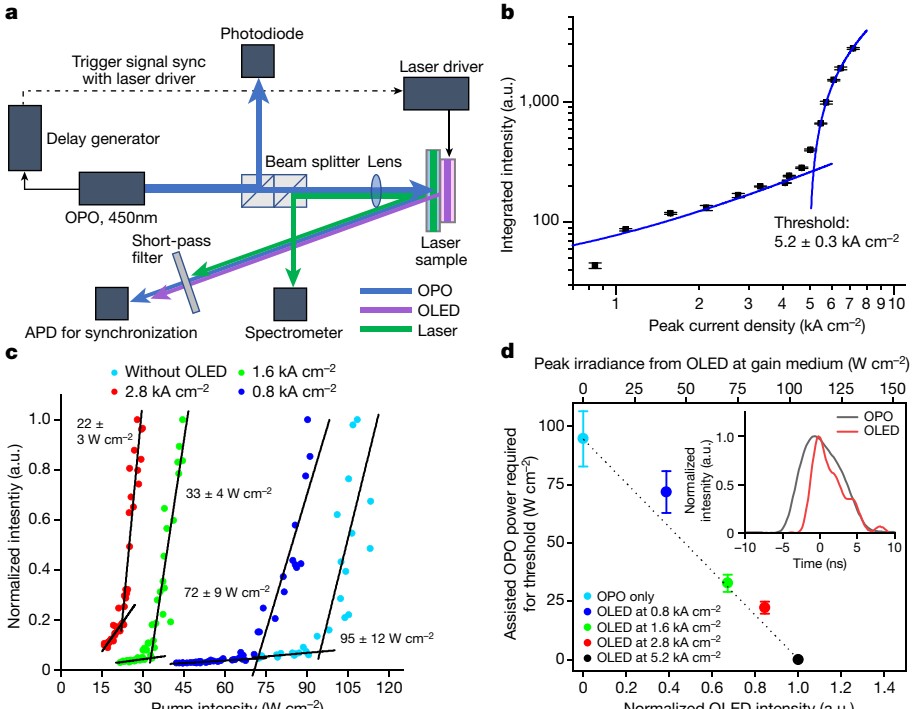

**Fig. 4 | Assisted optical pumping measurement of the electrically driven laser. a**, Schematic of the assisted pumping measurement set-up. **b**, Integrated laser output intensity as a function of peak current density. Error bars show the standard error of the measurement. The blue lines are linear fits to the data below and above the threshold. **c**, Normalized laser intensity as a function of optically pumped power density, while simultaneously providing sub-threshold electrical drive; each dataset is for a different peak current density. **d**, Assisted

OPO irradiance required to attain laser threshold as a function of OLED intensity (normalized to OLED output at threshold current density, bottom axis) and inferred peak irradiance from OLED at gain medium (top axis). Error bars show extended uncertainties of the measurement with a coverage factor of 2. Inset, time profiles of OPO and OLED emission and their relative timing during the measurement. a.u., arbitrary unit; APD, avalanche photodiode.

## Polymer laser design and operation

DFB polymer lasers were designed to achieve a low optically pumped threshold. We used a substructured grating, chosen to combine surface outcoupling with a high Q factor[29]. As mentioned above, the OLED output diverges markedly with increased distance from the laser, and its outcoupling efficiency is mainly limited by the refractive index of the contact layer. Thus, a thin top cladding layer with a high refractive index is required. We tested several top cladding layers, including a fluoropolymer (CYTOP, AGC Chemicals), an epoxy glue (NOA68, Norland Products) and PVPy. NOA68 and PVPy had a lower laser threshold than CYTOP because of their lower loss coefficient (Extended Data Fig. 4). PVPy was selected for the final device as it gives a much thinner and more controllable film thickness. We initially characterized the laser performance under optical pumping using an optical parametric oscillator (OPO). In Extended Data Fig. 5, the laser has a clear threshold of 92 W cm⁻² and a narrow emission peak at 541.5 nm when pumped above the threshold.

## Characterization of the integrated laser

There are four major criteria to recognize lasing: (1) threshold-like behaviour in both linewidth and output power; (2) narrow linewidth; (3) light output consisting of a beam; and (4) emission characteristic of the specific gain medium and resonator[38,39].

We tested our integrated laser by applying current pulses to the OLED. At a low peak current density of 2.0 kA cm⁻², the surface emission from the device showed a Bragg dip at 541 nm corresponding to the photonic stopband of the DFB laser (Fig. 3a). For higher injected current densities above 2.8 kA cm⁻², a narrow peak centred at 542.2 nm emerged on the long wavelength side of the photonic stopband corresponding to the

onset of laser emission. The integrated output intensity of the laser is plotted in Fig. 3b as a function of the peak current density. At low peak current densities, the intensity increases linearly before the slope of the emission changes abruptly at a threshold value of 2.8 kA cm⁻² to follow the S-shaped dependence characteristic of laser action. We identify this change as the threshold current density of 2.8 ± 0.2 kA cm⁻². The full width at half maximum of our laser for a current density of 4.9 kA cm⁻² is 0.09 nm, limited by the spectral resolution of the measurement system. The maximum output pulse energy was (1.5 ± 0.1) × 10⁻⁵ nJ, and we calculate the slope efficiency (peak optical power/peak input current) of the laser to be 2.1 ± 0.2 μW A⁻¹. The efficiency of the laser is currently limited by two factors: a large roll-off in OLED quantum efficiency under intense short-pulse operation and a low ratio of surface outcoupling to other losses in the DFB laser cavity. Further refinement of the cavity design and a better understanding of the dynamics of OLEDs under nanosecond-pulsed operation should each lead to notable improvements in laser efficiency in the future.

We measured the far-field spatial profile of the light emission with a charge-coupled device (CCD) camera (STF-8300M, SBIG Camera). Spatial profiles were taken below and above the threshold at different distances between the sample and the CCD camera (Extended Data Fig. 6). Figure 3c shows the beam profile 6 cm away from the sample. When the sample was operated below the threshold (at 2.0 kA cm⁻²), the image and the line profile only showed fluorescence from BBEHP-PPV. When the peak current density was increased to 4.9 kA cm⁻², a clear beam was observed. The double-lobe beam shape is consistent with the typical emission from a surface-emitting DFB laser with one-dimensional distributed feedback[40–42]. We extracted the 1/e² beam diameter from the beam image and line profile (Extended Data Fig. 7) and calculated the beam divergence of the narrow dimension to be 2.4 ± 0.2 mrad. We compared the beam profile under the electrical drive with the

emission optically pumped by the OPO. As shown in Fig. 3d, the optically pumped laser beam profile is very similar to the electrically driven device, further confirming that the emission from the electrically driven device is laser emission.

We also characterized the polarization of the electrically driven laser above threshold. As shown in Fig. 3b (inset) and Extended Data Fig. 7, the emission was strongly linearly polarized parallel to the grating groove direction, consistent with the emission typical of one-dimensional surface-emitting DFB lasers[40,42].

We measured the lifetime of the electrically driven laser by driving the device at 4.9 kA cm$^{-2}$ (1.7 times the threshold current density) at repetition rates of 10 Hz and 100 Hz. The results indicate that narrow laser emission was visible for $9.57 \times 10^4$ pulses (Extended Data Fig. 8), which corresponds to more than 2.5 h operation at 10 Hz. This is much longer than the 20 pulses observed in the report of indication of lasing under electrical pumping[25]. We separately tested the operational lifetime of the OLED under pulsed operation at an initial peak current density of 5.4 kA cm$^{-2}$ at 10 Hz and found an intensity drop of 10% of the initial intensity after $7.3 \times 10^4$ pulses. Thus, the lifetime of the laser under pulsed current operation may be attributed to the degradation of both the OLED and the polymer gain medium.

We also undertook measurements in which electrical and optical pumping could be combined[43,44] (Fig. 4a). These have three benefits: first, they enable direct comparison of electrical and optical pumping; second, for electrically driven structures that are not able to reach the threshold, it is possible to determine how close they are to the threshold; third, they enable an estimation of the irradiance transferred from the OLED to the gain medium. In Fig. 4, we used a second organic laser with a threshold for optical pumping of 95 W cm$^{-2}$ (much higher than the 47 W cm$^{-2}$ output into air measured from the OLED). This laser had a threshold under solely electrical excitation of 5.2 kA cm$^{-2}$. However, when driven at a lower current density, the laser can still reach the threshold when additionally excited optically (Fig. 4c,d). For example, when the OLED is operated at 2.8 kA cm$^{-2}$, an additional optical excitation of 22 W cm$^{-2}$ is needed to reach the threshold. These measurements show that the electrical driving of 5.2 kA cm$^{-2}$ is equivalent to an optical excitation of 95 W cm$^{-2}$. The measurements of electroluminescence in Fig. 2 show that for 5.2 kA cm$^{-2}$, 44 W cm$^{-2}$ is outcoupled into air. The higher equivalent power density of 95 W cm$^{-2}$ in the integrated device suggests that the coupling efficiency to the gain medium is enhanced by a factor of $2.4 \pm 0.3$ (after taking into account the differences in absorption between OPO and OLED excitation wavelengths), which is in good agreement with our calculated value of 2.3 for the coupling enhancement and demonstrates the benefit of our integrated device structure.

## Conclusion

In summary, we have demonstrated an integrated device approach to achieve electrically driven laser action in organic semiconductors, and so addressed an important challenge in organic optoelectronics. This approach overcomes the major difficulties commonly faced in the attempts of direct electrical injection of organic or hybrid perovskite lasers[25,43,45–48], while retaining the operational advantages. The observed threshold behaviour, spectral narrowing and polarized beam emitted from our devices provide strong evidence for lasing that is consistent with the properties of the gain medium and resonator used. The assisted optical pumping measurements are particularly helpful in quantifying the contribution of electrical driving to the laser threshold and will be a useful tool for future evaluation of other gain materials and structures to achieve electrically driven laser action.

Our approach to organic lasers requires the OLED to operate under exceptionally intense current injection to make a very fast organic optoelectronic device. We believe that the microscopic physics of OLEDs under such intense, short-pulse operation has not yet been fully explored. We anticipate that our work will stimulate future studies to understand the dynamics of organic semiconductors in this regime that could lead to notable improvements in device performance and enable further applications of ultrafast organic optoelectronics.

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

# Methods

## PNPN-substrates fabrication

Glass substrates were ultrasonically cleaned in detergent, deionized water, acetone and isopropanol followed by oxygen plasma treatment for 3 min. A self-assembled monolayer of trichloro(1H,1H,2H,2H-perfluorooctyl)silane was coated on substrates to enable easy peel-off after OLED fabrication. The substrates were then transferred to a nitrogen glovebox to coat the parylene and nanolaminate. Details of the method used can be found in ref. 30. In brief, two pairs of 1.5 µm of parylene and 50 nm of nanolaminate ($ZrO_2$ and $Al_2O_3$) were coated by a parylene coater (Labcoater 2, SCS) and an atomic layer deposition reactor (Savannah S200, Ultratech).

## PNPN-OLED fabrication

The as-prepared substrates were then transferred to the evaporator (EvoVac, Angstrom Engineering) for OLED fabrication. The OLED stack was coated on the PNPN-coated carrier glass by thermal evaporation in an evaporation chamber at a base pressure of $10^{-7}$ mbar. Four different custom-made shadow masks having a thickness of 100 µm were used. On the PNPN substrate, a 15-nm-thick layer of molybdenum trioxide ($MoO_3$, Merck) was evaporated through a shadow mask. Then, a 1.1 µm thick layer of aluminium was evaporated through another shadow mask (wiring electrode mask) to form the wiring electrode 1 for the cathode and the wiring electrode 2 for the anode (Fig. 1c). Aluminium was evaporated in a box heater with a crucible (EVCH5 and EVC5INTSPL01, Kurt J. Lesker) at rates of around 0.1 nm s$^{-1}$ for the initial 100 nm and then increasing gradually to be 2 nm s$^{-1}$ at maximum. Next, 1 nm gold as a wetting layer and 10 nm silver as a semi-transparent anode were evaporated through a bottom electrode mask. The $MoO_3$ and the gold layers were used to improve transmittance as well as sheet resistance[49]. Next, organic layer stacks with the organic layer mask were evaporated. The organic layer stacks consist of 30 nm of 1,1-bis(4-(N,N-di(p-tolyl)amino)phenyl)cyclohexane p-doped with 2,2′-(perfluoronaphthalene-2,6-diylidene)dimalononitrile at 4 wt% as a hole-transport layer, 2.5 nm of 9,10-dihydro-9,9-dimethyl-10-(9-phenyl-9H-carbazol-3-yl)-acridine and 7.5 nm of 4,4′,4′′-tris(carbazole-9-yl) triphenylamine as electron-blocking layers, 20 nm of TSBF as emission layer, 10 nm of 1,3,5-tris(1-phenyl-1H-benzimidazol-2-yl)benzene as a hole-blocking layer, and 30 nm of caesium-doped 4,7-diphenyl-1,10-phenanthroline as an n-doped electron-transport layer. After the evaporation of the hole-transport layer, the evaporation chamber was vented to swap and introduce materials for the later evaporation. Around 5 µm of aluminium was evaporated to form the cathode and wiring electrodes 3 for the anode, with another shadow mask (top electrode mask). The Al evaporation was split into three sections to prevent the OLED from warming up during the evaporation. The top surface of the evaporated aluminium became cloudy for thicknesses of more than 1 µm and did not show metallic reflection. However, the bottom surface remained mirror-like and the resistance decreased as the thickness increased.

On top of the aluminium, a 100-nm buffer layer of $MoO_3$ was evaporated through the organics masks and then the OLED was encapsulated with a 100-µm-thick flexible glass (MNJ-300-010 W, MENZEL0GLASER) and a pre-baked epoxy glue (Norland Products, Norland Optical Adhesive 68) under nitrogen conditions. The flexible glass was attached to prevent shrinkage of OLEDs after peeling off from the carrier glass (Extended Data Fig. 9). Before use, the glue was baked at 100 °C for several hours and stored in the glovebox at least for a week. The $MoO_3$ layer and the baked glue were used to prevent damage to the OLED-active area when we attach the glass.

The organic materials for OLED fabrication were purchased from Luminescence Technology and used as received. The size of the OLED active-area was estimated by measuring the area where the cathode and anode overlapped under a microscope (ECLIPSE LV100ND, Nikon). Widths of the OLEDs were slightly different depending on the samples,

120–150 µm, whereas the lengths were similar (1 mm), probably because of the contact of the top shadow mask and the substrate. Thus, the sizes of all samples were measured and used to estimate the performance. PNPN-OLEDs with similar radiant exitance were made with a yield of around 60%.

By using a similar procedure, films of the emission layer, TSBF, on glass or fused silica were made. The films on glass were used to measure photoluminescence spectra and encapsulated with epoxy glue and a 1.1-mm-thick glass lid (Shanghai Amerina Optoelectronic) under nitrogen conditions.

## Driving OLEDs under pulse mode and their characterization

The OLEDs were driven with a laser driver (EPC9150, EPC) to produce intense light output by injecting a high current while suppressing Joule heating and causing breakdown. Reduction of stray inductance is important to obtain a short pulse using high current[50]. A custom-made printed circuit board (PCB) was used to make contact with the OLEDs (Extended Data Fig. 10a–c). The board and the driver were wired with a strip line consisting of Kapton tape sandwiched between two copper tapes of width 1.5 cm on both sides. On the PCB, three 33-Ω resistors were connected in parallel to the OLED connection to bypass the current after turning off and to remove a high voltage that would otherwise remain for more than 120 ns. The current output from the driver was determined by measuring the output voltage of the built-in resistor on the board with a 500-MHz bandwidth oscilloscope (TDS 3052C, Tektronix) with a conversion constant of 120 A V$^{-1}$ reported by EPC. The measured current included the current passing through the bypass resistors. Thus, to determine the current passing through the OLED, the current passing through the resistors measured without any OLEDs was subtracted from the current measured with the OLEDs. Extended Data Fig. 10d shows the output current of the driver on the PCB in shorted and open circuit conditions as a function of the voltage applied to the driver. In both conditions, the peak current increased linearly with the driver voltage. When shorted, the peak current was 2.4 A at a driving voltage of 20 V. Ringing current (Extended Data Fig. 10e) was observed with a positive-peak-to-positive-peak period of 13.5 ns. The oscillation is attributed to a resonance of parasitic inductance and capacitance.

Temporal profiles of electroluminescence from the OLEDs were measured with a fast avalanche photodiode (APD) having a bandwidth of 400 MHz (APD430A2, Thorlabs) and an oscilloscope. A pair of lenses were used to image the electroluminescence onto the photodiode. The temporal response, measured with a 100-fs pulse-width laser, had a rise time of 1.2 ns and a fall time of 0.7 ns.

The peak radiant exitance from the OLED was estimated by measuring the pulse energy from the OLED with a calibrated energy meter (J3S-10, Coherent) placed within 1 mm of the OLED and by measuring the time profile of the optical pulse. The emission spectra of the PNPN-OLEDs were measured with a calibrated fibre-coupled spectrometer (USB4000, Ocean Optics) by placing the fibre normal to the substrate surface. EQE was estimated considering peak irradiance, peak current density and electroluminescence spectra.

## Characterization of film samples

BBEHP-PPV was synthesized following our previously reported method[36]. The absorptivity spectrum of a 230-nm-thick BBEHP-PPV film was inferred from the transmission spectrum measured using a variable-angle spectroscopy ellipsometer (J. A. Woollam, M-2000 Ellipsometer). Optical constants of materials used in the integrated structure were measured with the same ellipsometer. Photoluminescence spectra of the TSBF film were measured by using a fluorimeter (Edinburgh Instruments, FLS980).

To measure the waveguide loss, the pump beam was focused into a narrow stripe shape (2.3 mm × 130 µm) using a cylindrical lens. The end of the stripe was positioned near the edge of the waveguide. The pump stripe was moved away from the edge of the film and the

emission from the edge was collected by a fibre-coupled CCD spectrometer. The emission intensity from the edge was fitted by $I = I_0\exp(-\alpha x)$, where $I_0$ is a constant intensity, $x$ is the distance of the stripe from the edge of the film and $\alpha$ is the waveguide loss coefficient.

## Calculation of outcoupling efficiency

Outcoupling efficiencies of OLEDs were simulated by treating the emission dipole as a forced damped harmonic oscillator embedded in a thin-film stack[51–53]. The simulation was conducted within a wavelength range of 300–800 nm with a 1-nm step and, at each wavelength, dissipation powers were calculated for normalized in-plane wavevectors from 0.0 to 2.0 with a step size of 0.002. This maximum in-plane wavevector was chosen to ensure that the calculation captures the dissipation into all evanescent modes in the device and is a wider range than previously used in related OLED calculations[51]. We confirmed that the evanescent surface plasmon polariton (SPP) mode of the device has negligible power beyond this range. In the model, the PNPN substrate was included in the OLED stack; the optical constants of parylene and nanolamination layers were obtained from ref. 30. The emission dipole anisotropy factor ($\alpha$) was considered for the calculation. The $\alpha$ of TSBF was determined to be 0.082 from the refractive index spectra of TSBF by using the following equation,

$$\alpha = \frac{2}{3} \times \left( \frac{k_e - k_o}{k_e + 2k_o} \right) + \frac{1}{3}$$

where $k_o$ and $k_e$ are the ordinary and extraordinary extinction coefficients at the longest peak wavelength. The emitter dipoles were assumed to be localized at the interface of the 4,4′,4-tris(carbazole-9-yl) triphenylamine and TSBF layers. The effect of different dipole positions on outcoupling efficiency to air or to PVPy was calculated; the fractional error from any uncertainty in the emission zone profile is estimated to be no more than 10%.

## Calculation of irradiance distribution

The irradiance distribution in a plane parallel to the OLED substrate at a distance ($z$) was simulated by treating the OLED as an assembly of many point light sources, calculating light to a point receiver at an in-plane position ($x$, $y$) from each light source, and summing all contributions. The origin was set at the centre of the OLED-active area, and only positive $x$ and $y$ directions were calculated considering the symmetry of the OLED along each axis. The irradiance at a position ($x$, $y$, $z$) can be calculated as the total energy per unit OLED area received at the receiver as follows:

$$I(x, y, z) = \sum_i R(\theta_i) \times \frac{\cos(\theta_i)}{r_i^2} \times \Delta S$$

where $R(\theta_i)$ is the angular distribution of OLED emission, $r_i$ and $\theta_i$ are the distance and polar angle between the $i$th section of the OLED of area $\Delta S$, and a receiver located at ($x$, $y$, $z$).

$R(\theta_i)$ is related to the total output power per unit area from OLED ($P_{OLED}$) by integration over a hemisphere as follows:

$$P_{OLED} = 2\pi \int_0^{\frac{\pi}{2}} R(\theta) \times \sin(\theta)\, d\theta$$

In the simulation, the OLED was divided into 1 μm × 1 μm sections and $P_{OLED}$ was normalized to a value of 1. The OLED emission pattern was calculated using the same simulation as the outcoupling efficiency. We simulated the emission pattern of the PNPN-OLED emitting to parylene.

## Literature survey of device performance and details of data collection

The data used to plot Fig. 2d and Extended Data Fig. 3 are provided in the Supplementary Information and refs. 25,28,31,32,47,54–106.

We collated the performance characteristics of two-terminal vertical organic devices that achieve a high current density of 10 A cm$^{-2}$ or more. We included information on OLEDs and unipolar devices. In some studies, the peak wavelengths were not reported. In such cases, the peak emission wavelengths were used from other publications reporting either electroluminescence spectra or photoluminescence spectra of the same emission layers. In other cases, the OLED emission characteristics were reported as only luminance, radiant exitance or EQE. The unreported values were estimated from the available data by assuming that the OLEDs were monochromatic at their peak wavelength with Lambertian emission. When both EQE or radiant exitance and luminance were reported, EQE or radiant exitance were chosen instead of luminance. This is because the values calculated from luminance may vary markedly depending on the emitter spectrum.

## Laser sample fabrication

Laser gratings were fabricated by nanoimprint lithography using an EVG 620 photomask aligner with customized tooling. First, a 350-nm bottom cladding layer was spin-coated from 150 mg ml$^{-1}$ silsesquioxane-based polymer (H-SiO$x$, HSQ from EM Resist) in methyl isobutyl ketone solution onto a glass substrate, and then annealed at 340 °C for 30 min. This cladding layer was then covered with a spin-coated layer of UVCur21 nanoimprint resist (micro resist technology). A perfluoro-polyether nanoimprint stamp was replicated from a silicon master structure (cured by ultraviolet light exposure) and then used to nanoimprint the grating structure into the UVCur21 layer. The nanoimprinted grating was then used as an etching mask in reactive ion etching to transfer the grating pattern to the bottom cladding layer. The depth of the grating was controlled by varying the etching time. An oxygen plasma was applied to remove the residual resist after etching. The etched gratings were treated by oxygen plasma at low power for 20 s before spin-coating with BBEHP-PPV from a 16-mg ml$^{-1}$ toluene (anhydrous, 99.8%, Sigma Aldrich) solution. A top cladding layer of poly(vinyl-pyrrolidone) (PVPy; PVP40 from Sigma Aldrich) was spin-coated on top of the BBEHP-PPV film from a 280-mg ml$^{-1}$ 1-butanol (anhydrous, 99.8%, Sigma Aldrich) solution. Finally, a 1.5-μm thick parylene layer was deposited by a parylene coater as the contact layer. Lasers with similar thresholds were made with a yield of around 60%.

To integrate the PNPN-OLED and the BBEHP-PPV laser, the PNPN-OLED was first peeled off the glass carrier, and the OLED-active area was manually aligned to be located within the grating region of the organic laser, with the longer axis of the OLED aligned perpendicular to the grating grooves (Fig. 1c). A glue gun (BOSCH, PKP 18E) was used to fix the alignment of the OLED and the laser by applying a small bead of glue at the substrate corners. To ensure good physical contact between the OLED and the laser, a small elastomer cushion of area 2 mm × 1 mm and thickness 1 mm was attached with glue (KISS Powerflex brush-on nail glue) on top of the flexible glass to cover the whole OLED-active area, and then gentle mechanical pressure was applied to the integrated device stack using a custom-made clamp. The integration was conducted in a clean room to avoid the inclusion of particles between the OLED and the organic laser, which was found to be important for stable operation. Integration of the OLED and the BBEHP-PPV laser is an important step in the fabrication that can affect the laser threshold current density. We tested 14 devices in this study and three showed narrow emission (with a yield of around 20%). We tested the spatial output of two devices, and both showed a similar beam-like spatial distribution of the output light (Fig. 3). The low yield is probably because of the manual application of pressure in the integration, which was not consistently controlled. By improving the assembly procedure, the reproducibility may be improved.

## Optically pumped laser characterization

In optically pumped laser measurements, the samples were pumped with nanosecond pulses by an OPO (Horizon OPO, Continuum) at

450 nm with a repetition rate of 20 Hz. The intensity of the pump laser was controlled by a variable neutral-density wheel filter, and the pulse energy was measured with a calibrated photodiode (Thorlabs). The beam was focused into a circular spot of about 1 mm diameter with a quasi-top-hat intensity profile. The output emission was collected by a fibre-coupled CCD spectrometer (Andor DV420-OE).

### Electrically driven laser characterization

In electrically driven laser measurements, the sample was driven by a pulsed current laser driver (EPC9150, EPC) at a repetition rate of 10 Hz. The output emission was collected with the fibre-coupled CCD spectrometer. The output beam profile of the laser sample was captured with a CCD camera (SBIG Camera STF-8300M) at different distances. The output pulse energy and slope efficiency of the electrically driven laser were determined by calibrating the response of the CCD camera to a calibrated energy meter (J3S-10, Coherent); the organic laser pulse duration was measured to be 2.3 ± 0.1 ns using a silicon avalanche photodiode (APD430A2, Thorlabs). The polarization of the laser beam was measured by inserting a linear polarizer (LPVISE100-A, Thorlabs) at different polarization angles between the sample and the spectrometer. For lifetime measurements, the sample was driven at a fixed current above the threshold, and a time series of spectra was recorded using the CCD spectrometer in kinetics mode.

### Assisted pumping laser measurements and estimation of corresponding OLED light output

For assisted optical pumping laser measurements, the pulsed laser driver was triggered by a synchronized signal from the OPO trigger output. To precisely synchronize the pulses, the temporal emission profiles of the OLED and OPO were monitored using an avalanche photodiode (APD430A2, Thorlabs). A custom-made delay generator was used to tune the delay between the OPO and OLED pulses and maximize the overall peak pump irradiance by overlapping the peaks of OPO and OLED emission. A short pass filter with a cut-off wavelength of 500 nm (FES0500, Thorlabs) was inserted before the APD to remove any emission from the BBEHP-PPV laser. To quantify the contribution from electrical driving, the OPO-pumped threshold was first measured without current injection. The integrated sample was then driven at a fixed peak current density, while simultaneously pumped by the synchronized OPO pulse as the OPO power density was varied. The optically pumped laser thresholds under different OLED peak current densities were measured. The reduction in the optically pumped threshold for each peak current density was attributed to the contribution of electrical driving. The electroluminescence power density at the gain medium was then inferred from the change in OPO-pump threshold, taking into account the different emission spectra of the PNPN-OLED and the OPO, and the corresponding ratio of absorptivity of BBEHP-PPV for the two pump sources. Normalized OLED intensities in Fig. 4d were obtained by taking the ratio of the peak radiant exitance of the OLED at each drive current to radiant exitance measured at 5.2 kA cm$^{-2}$. The electroluminescence power density needed to create an equivalent exciton density in the BBEHP-PPV layer was thereby estimated to be around 1.1 times greater than that of OPO.

### Measurement uncertainties in OLED and laser characterization

Uncertainties in our measurements are expressed as extended uncertainties for an interval of 95% confidence (coverage factor $k = 2$). These take account of combined uncertainties in the calibration of the energy meter and the scale of the microscope used for the OLED size measurements, the reading resolution and time resolution of the oscilloscope, pixel-to-pixel variations of OLED light output and variations in repeated measurements, as well as uncertainties in the linear fitting of laser output to estimate the lasing threshold. The uncertainties in radiant exitance, EQE and the laser threshold under optical pumping include

a 10% calibration uncertainty of the energy meter. The uncertainty in threshold current density is mainly because of the calibration scale in the microscope used to measure the size of the OLED.

## Data availability

The research data that support the findings of this study are available[107]. Source data are provided with this paper.

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

**Acknowledgements** We thank the Engineering and Physical Sciences Research Council of the UK for the financial support from grants EP/R035164/1, EP/R03480X/1 and EP/L017008/1. J.G. thanks the China Scholarship Council (grant no. 201806100005) for financial support. We thank M. Ross for work on the development of the power supply unit for the laser driver, together with the clean room technician team, and electrical and mechanical workshops in the School of Physics and Astronomy at the University of St Andrews. To meet the institutional and research funder open access requirements, the accepted manuscript shall be open access under a Creative Commons Attribution (CC BY) reuse licence with zero embargo.

**Author contributions** I.D.W.S., G.A.T. and K.Y. conceived of the project and I.D.W.S. and G.A.T. supervised the research. K.Y. led the development of high-brightness OLEDs, J.G. made the lasers, and both K.Y. and J.G. worked together to integrate the devices and characterize their operation. A.L.K. and P.J.S. synthesized and provided the BBEHP-PPV polymer. All authors wrote sections of the paper and edited the entire paper.

**Competing interests** The authors have filed a patent based on the results in this paper.

**Additional information**
**Correspondence and requests for materials** should be addressed to Graham A. Turnbull or Ifor D. W. Samuel.

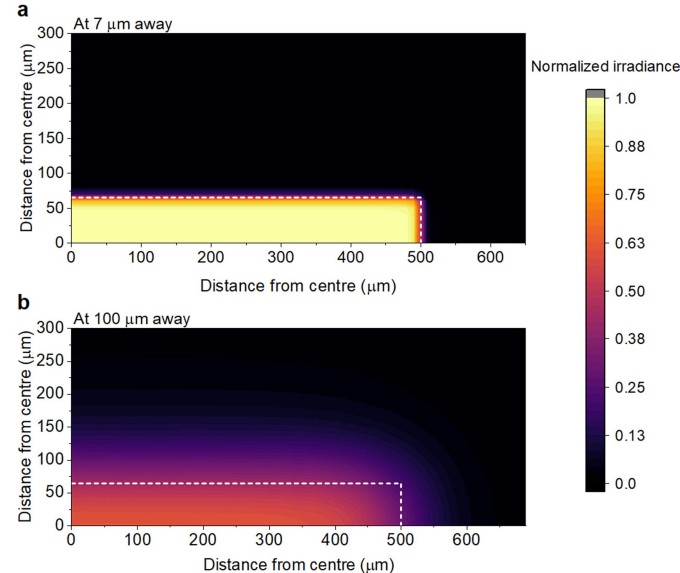

**Extended Data Fig. 1 | Calculated distribution of irradiance from a PNPN-OLED of width 130 μm and length 1 mm.** Irradiance calculated at distances of **a**, 7 μm and **b**, 100 μm. The origin was set at the centre of the OLED active area and only positive directions were calculated due to the symmetry of the OLED along each axis. Hence only a quarter of the OLED is shown. White broken lines show the active area of the OLED.

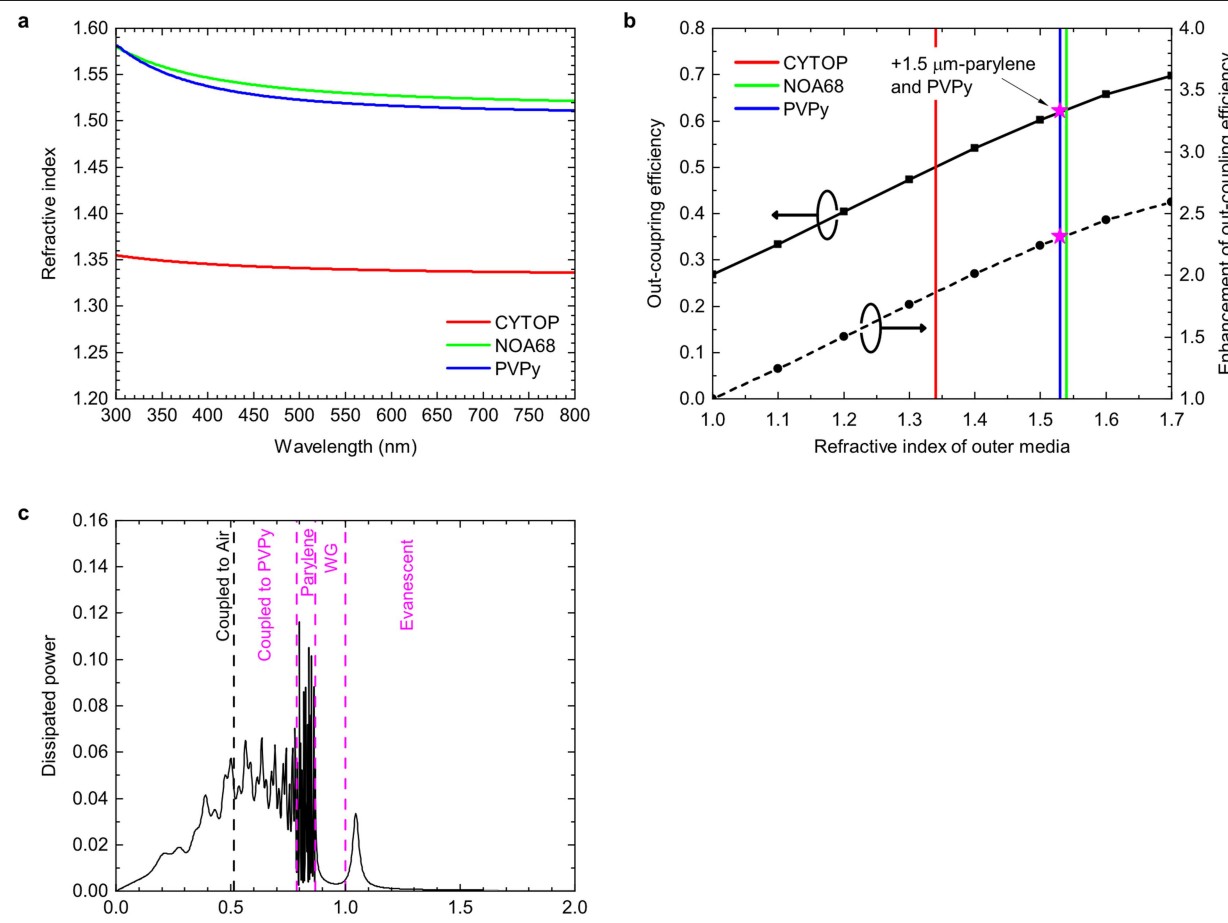

**Extended Data Fig. 2 | Outcoupling efficiency simulation based on cladding layers with different refractive index. a**, Refractive index spectra of different candidate materials for the cladding layer. **b**, Calculated outcoupling efficiency of the PNPN-OLEDs as a function of the refractive index of the final outcoupling medium, and the enhancement of the outcoupling efficiency compared with emission into air. In part **b**, stars show the values calculated for the PNPN-OLED with an additional 1.5 μm parylene and PVPy as the coupling layer and vertical lines show refractive indexes of the cladding layers at 430 nm. **c**, Calculated dissipation spectrum at 430 nm as a function of the normalized in-plane wavevector. The regions separated by the dashed lines indicate the regions of the device into which the emission is trapped (and dissipated). The losses in the structure (integrated over the full emission spectrum) are 8% in evanescent modes, 14% absorption in the semi-transparent electrode, 16% in waveguide modes and the parylene layer, leaving 62% within the light cone entering the PVPy cladding of the laser waveguide. We assume that in the integrated structure all of this light entering the PVPy layer may be used to pump the polymer laser. For the case of a separate OLED and organic laser, however, only the light within the air light cone (in-plane wavevector < 0.5) may be utilised to pump the laser.

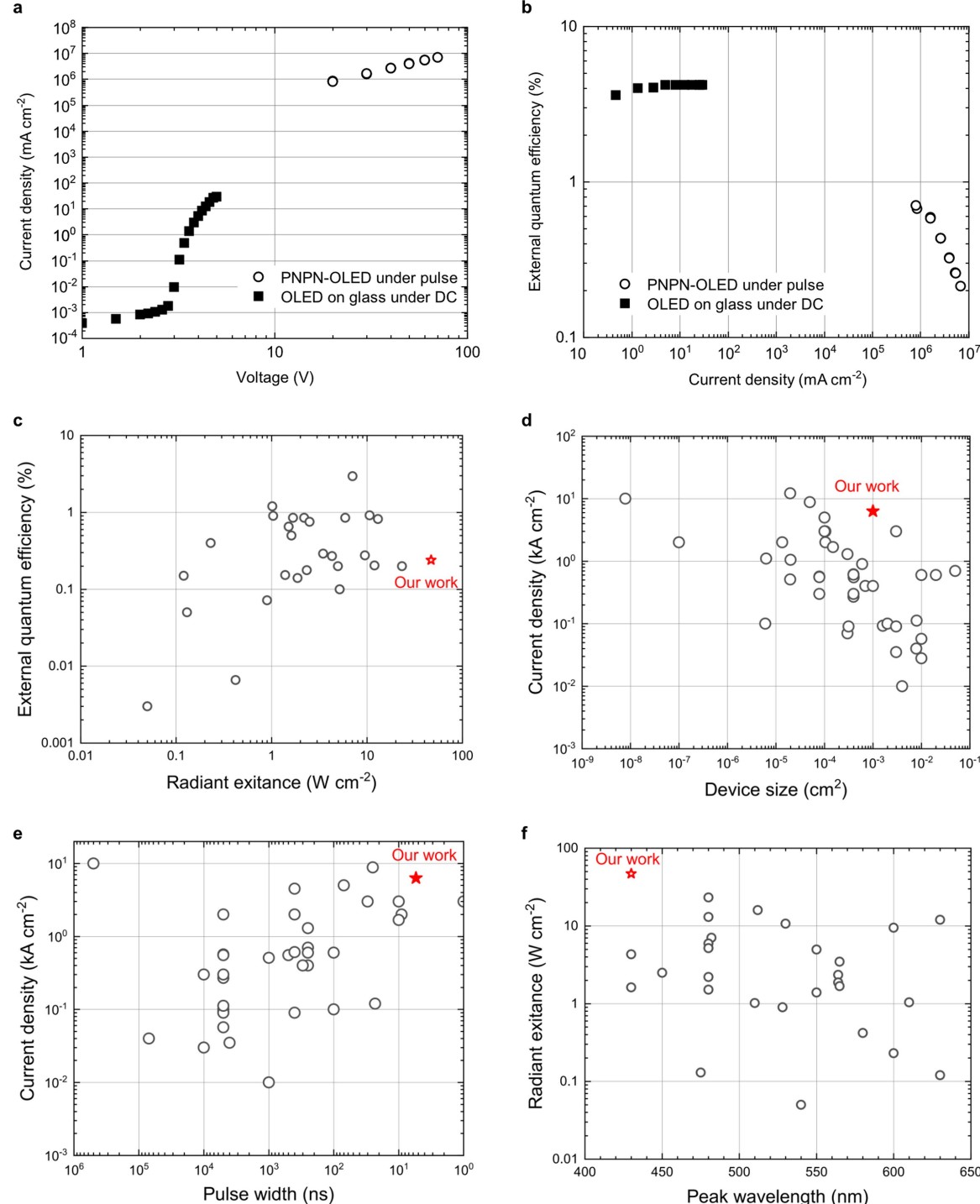

**Extended Data Fig. 3 | Performance comparison of reported OLED-like devices at high current density.** See also Supplementary Information table 1. Example DC (squares) and pulsed (circles) characteristics of the OLED used in the present study: **a**, voltage-current density and **b**, EQE-current density.

Comparisons of OLED performance with reported two-terminal vertical organic devices: **c**, reported external quantum efficiencies as a function of radiant exitance; **d**, Current density as a function of device size; **e**, current density as a function of pulse width; **f**, Radiant exitance as a function of peak wavelength.

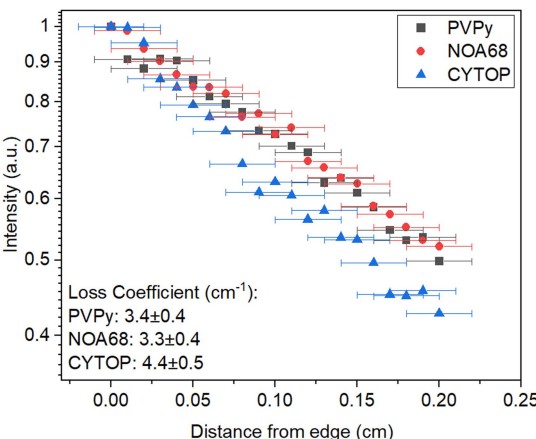

**Extended Data Fig. 4 | Waveguide loss measurement of BBEHP-PPV films with different top cladding layer.** Amplified spontaneous emission intensity from the edge of the waveguide as a function of the distance between the edge and the pump beam. An uncertainty of 0.02 cm in distance from edge was included in our measurement and fitting.

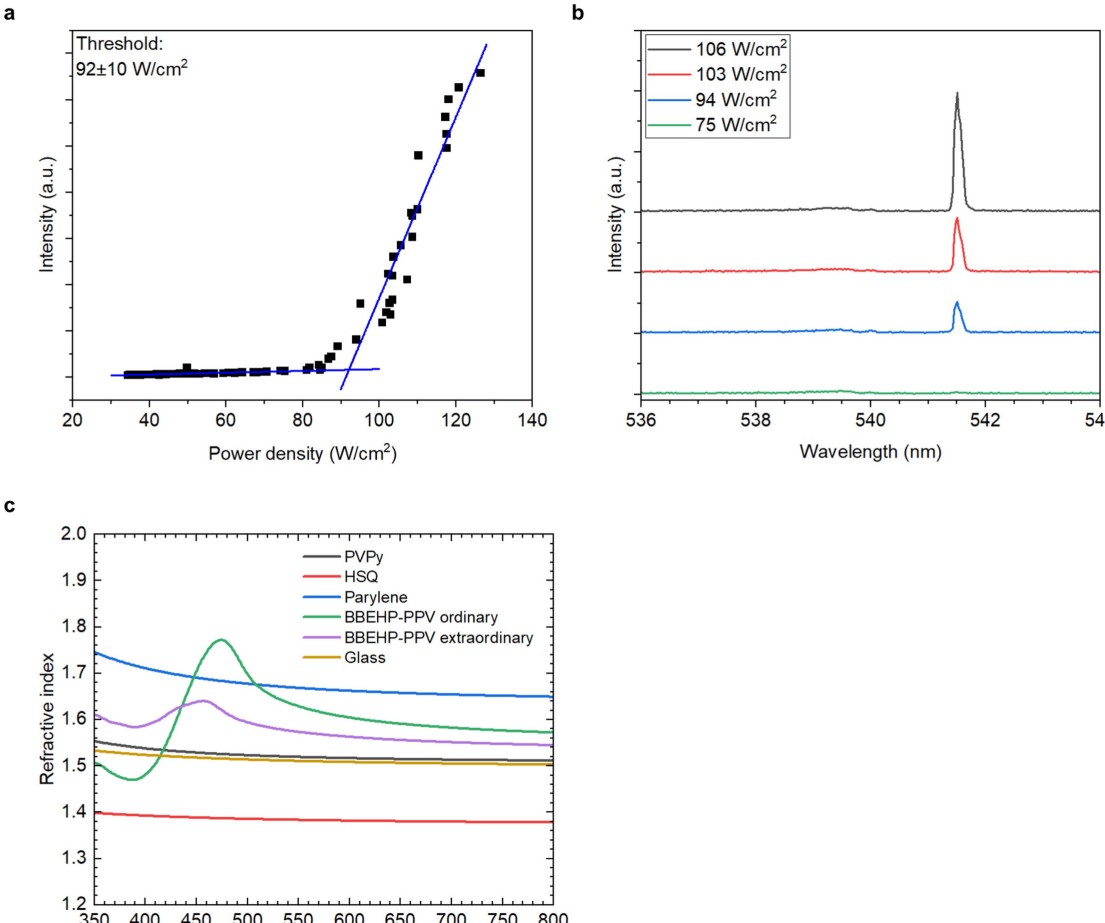

**Extended Data Fig. 5 | Characterization of the laser sample under optical pumping operation. a**, Output intensity as a function of OPO pump power density. The blue lines are linear fits to the data below and above threshold. **b**, Evolution of emission spectra under different pump power density below and above threshold (offset for clarity). **c**, Refractive index spectra of the different layers in the organic semiconductor laser.

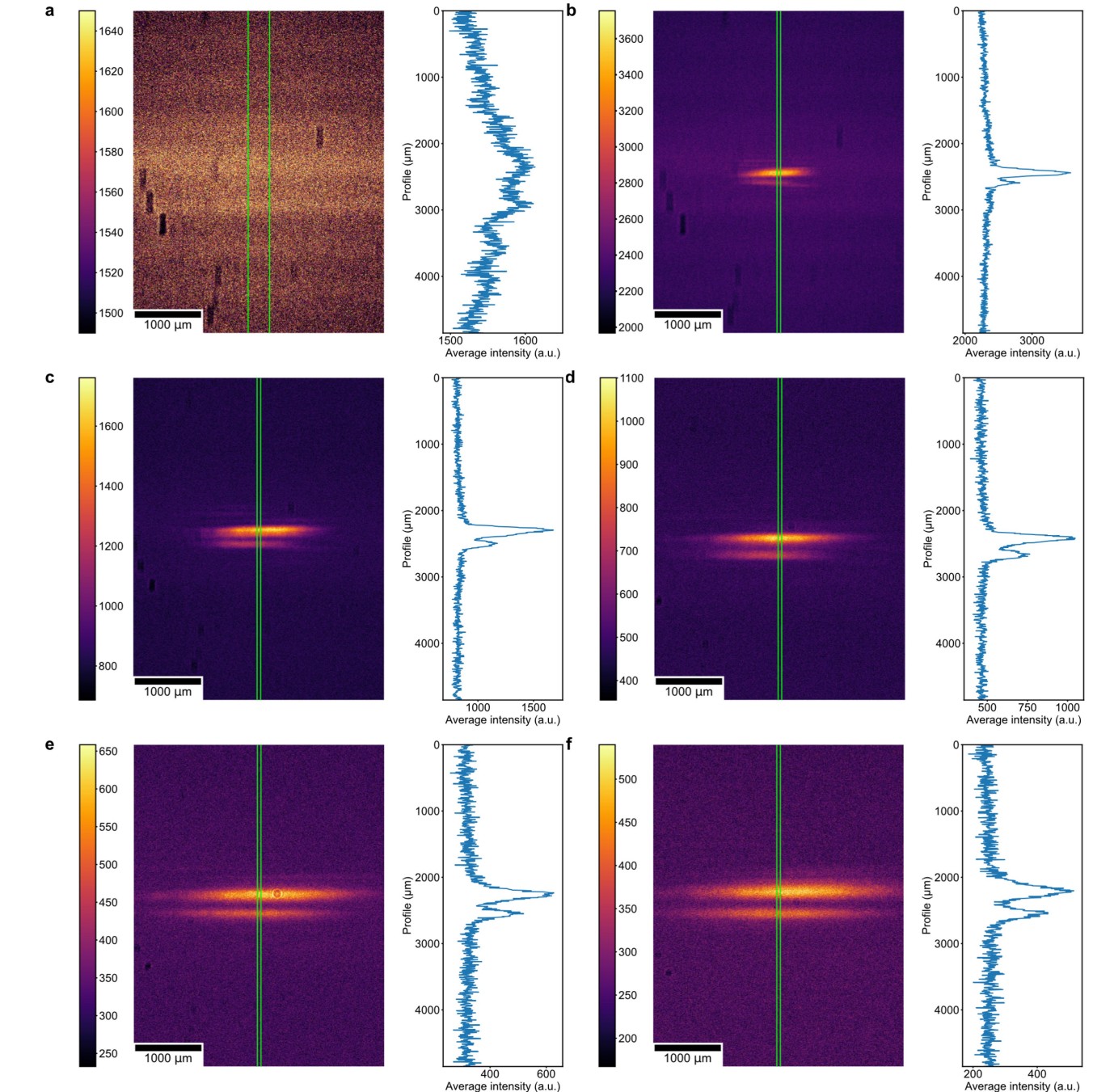

**Extended Data Fig. 6 | Images of the laser beam and line profiles at different distances between laser and CCD camera. a**, At 2.2 cm distance from the camara and pumped below threshold. At **b**, 2.2 cm, **c**, 4 cm, **d**, 6 cm, **e**, 8 cm, and **f**, 10 cm distance from the camera and pumped above threshold.

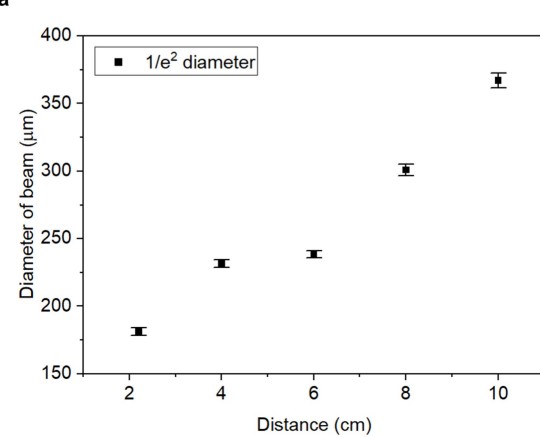

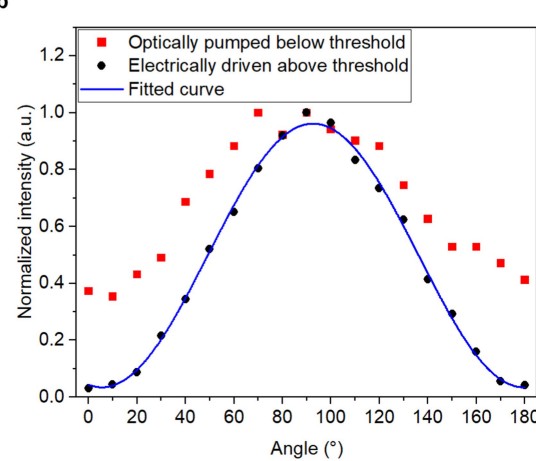

**Extended Data Fig. 7 | Characterization of the electrically driven laser beam.**
**a**, Beam diameter extracted from the beam images and line profiles. Beam
diameter ($1/e^2$, measured across the narrow dimension) of the higher intensity
lobe in Extended Data Fig. 6b–f, as a function of the distance between the
laser and the CCD camera. Error bars are standard error of the fitting. **b**, The
normalized intensity as a function of the linear polarizer angle when the
sample is electrically driven above threshold, and optically pumped 20% below
threshold, the blue curve is a least squares fit to a $\sin^2(\theta)$ function; 90° is defined
as the polarisation parallel to the grating groove direction.

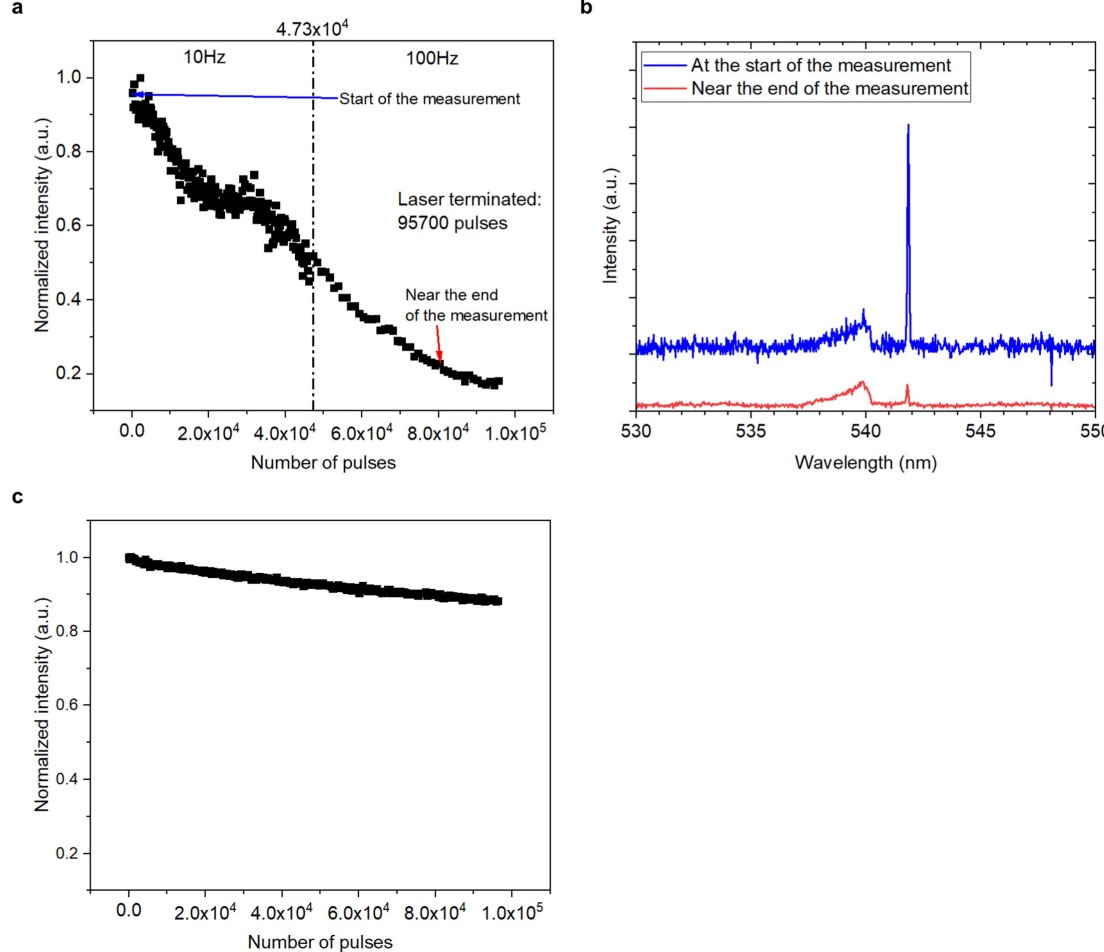

**Extended Data Fig. 8 | Operational lifetime of the electrically driven laser. a**, Normalized laser peak intensity as a function of number of current pulses. **b**, Laser spectrum recorded at the start, and near the end, of the lifetime measurement. **c**, Normalized OLED peak intensity as a function of number of current pulses.

**a**

Without OLED structure

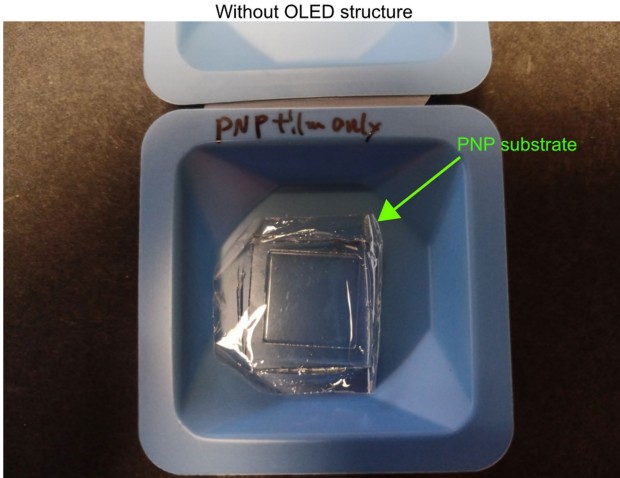

PNP substrate

**b**

Without flexible glass

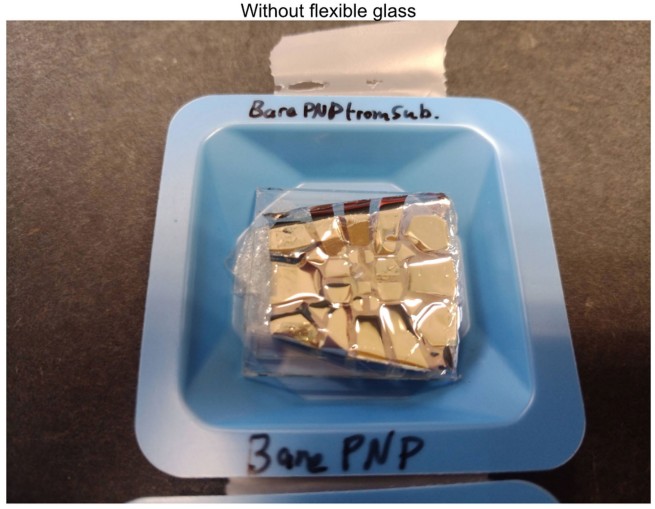

**c**

With flexible glass

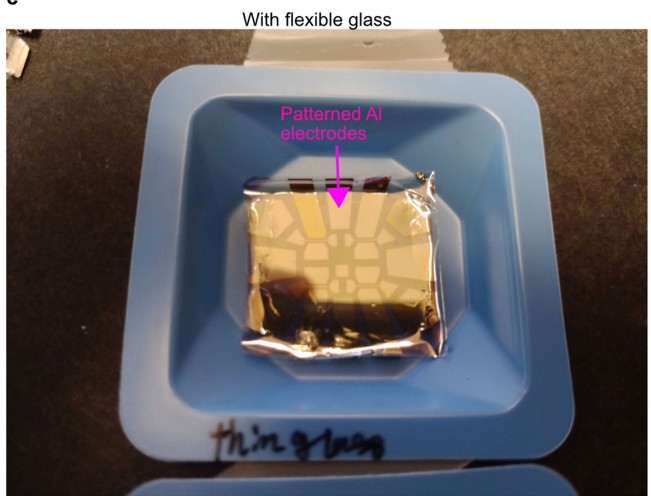

Patterned Al electrodes

**Extended Data Fig. 9 | Photos of PNP-substrates with different layers after removal from the carrier glasses. a**, A three-layer PNP substrate; **b**, PNP substrate with OLED-like test structure, Ag(20 nm)/ *N*,*N*′-Di(1-naphthyl)-*N*,*N*′-diphenyl-(1,1′-biphenyl)-4,4′-diamin (NPB, 40 nm)/Al (2.45 μm); **c**, the same OLED-like structure with top layer of flexible glass. The PNP substrate itself remains flat even after removal from the carrier glass, but the OLED-like structure shows crumpling along edges of the patterned Al films. This can be supressed by adding the flexible glass substrate on top of the OLED structure before removal from the carrier glass.

**a**

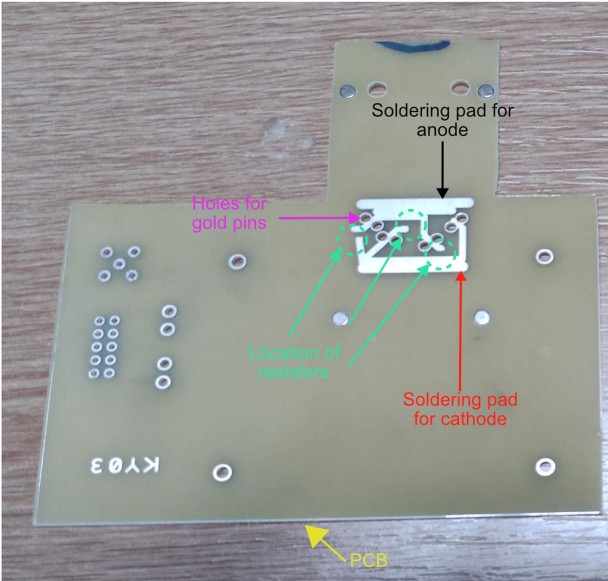

**b**

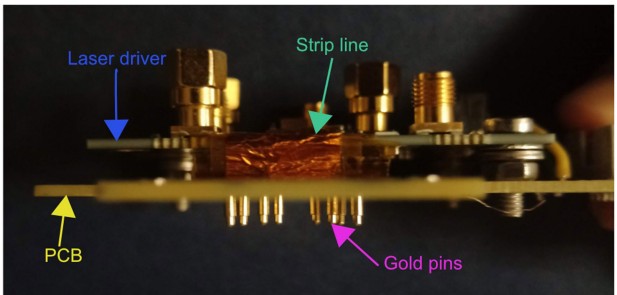

**c**

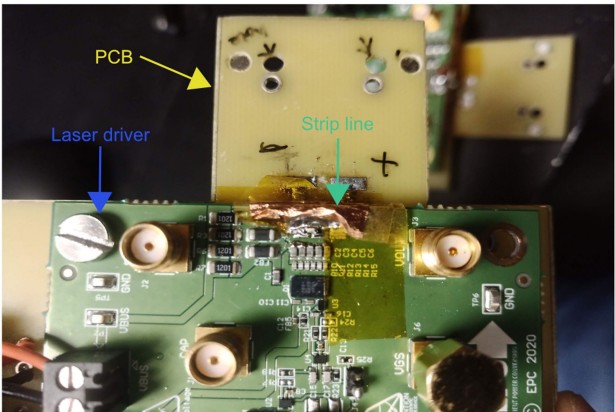

**d**

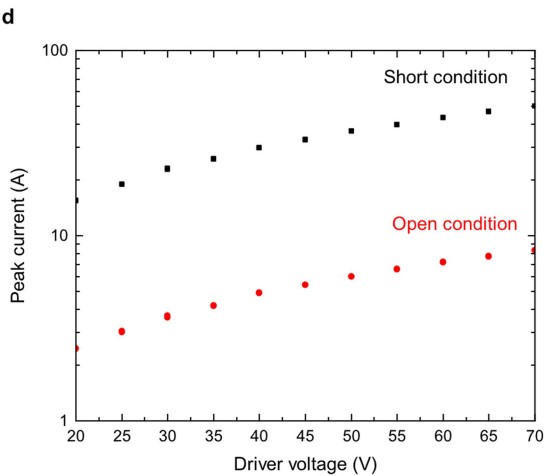

**e**

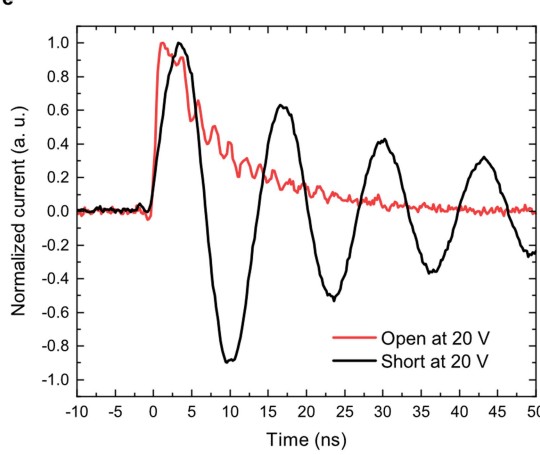

**Extended Data Fig. 10 | Laser driver images and characterization. a**, Custommade PCB for the laser driver and **b**, cross-sectional and **c**, top view of the PCB with the laser driver wired with the strip line and with gold pins. **d**, Peak output current of the driver of the OLED measurement holder in open and short circuit conditions as a function of laser driver voltage. **e**, Time profile of output current in open and short circuit conditions at a driver voltage of 20 V.