## [Peer Review File · Nature]

Manuscript Title: Electrically driven organic laser using integrated OLED pumping

Reviewer Comments & Author Rebuttals

Reviewer Reports on the Initial Version:

Referees' comments:

Referee #1 (Remarks to the Author):

The manuscript by Yoshida and co-workers describes the realization of an integrated device allowing for electrically-driven organic lasing in an all-organic structure consisting of a thin OLED with doped injection layers and an organic (BBEHP-PPV) DFB laser structure. Electrically-injected organic laser has long been a holy grail of organic semiconductor research, but has been hampered by various effects such as the build up of long-lived triplets, polarons and poor thermal stability of most organic semiconductors. The first "real" evidence for electrically-driven organic laser was recently published in Ref. 26. However, devices in that work were very short-lived (10-50 pulses) and yield was low. In this submission, Yoshida effectively bypass the many difficulties related to electrical injection by optically pumping the organic laser with an OLED integrated in close proximity. While neither the OLED nor the organic laser structure show significant advances over previous reports, this is quite an engineering feat, which required solving some integration issues, with an end result that I believe is appropriate for Nature. While in some sense, this is very similar to previous work from the Samuel group where organic lasers were integrated with inorganic LEDs, the fact that this would also be achievable with an OLED was far from obvious. Whether such a device will be of practical use remains to be seen.

The work is relatively straightforward and the numbers add up in such a way that I have confidence that the results are correct. Still, I do have a number of recommendations and questions. While I did not have any major problems following the manuscript, I think that it could be written more clearly (and directly) in a way that is a better fit for the readership of Nature.

Recommendations:

-The title is succinct, but I do not feel it correctly reflects the content of the work. One could also have used the same title for previous work from the same group using GaN LED or laser diode pumping. I suggest highlighting the fact that the structure is all organic, while also avoiding the insinuation that this is an organic laser diode.

-The complete absence of uncertainties (especially given the topic) in the manuscript is somewhat shocking. Many values are given to 3 significant digits, with no uncertainty and not a single error bar can be found on the plots. Type A and B uncertainties on the various measured quantities should be reported/combined using statistical analysis and calibration reports and details of how uncertainties have been calculated should be reported.

-The most basic measures of laser performance are the slope efficiency and wall-plug efficiency. There are no actual values on the y axis for any of the electrically-driven laser plots. This should be fixed.

-I do not think it is appropriate for the authors to use 7 lines of their introduction to denigrate Ref. 26, in what seems like an attempt to increase the importance of their own work. I believe that simply highlighting that the diodes in the one report on OSLEDs are extremely short-lived (tens of pulses) and that yield was low is sufficient.

-It would be nice to include photostability data for the OLED to accompany Ext. Data Figure 8.

-The use of radiant exitance for the output intensity is somewhat esoteric. In the 1000s of papers on LEDs since the 1960s, I do not ever recall seeing its use for the EL Intensity (despite the radiometric meaning being correct). In the field of infrared LEDs, "emittance" is sometimes used, but is also less common than Intensity/Irradiance.

-While the peak brightness ($48\text{W}/\text{cm}^2$ at $7\text{kA}/\text{cm}^2$) achieved from the OLED is quite good, the performance seems on par with other reports, which show peak brightnesses of around $10\text{W}/\text{cm}^2$ at $0.5\text{-}1\text{kA}/\text{cm}^2$. The only difference is that the OLEDs in this manuscript were driven a little bit harder (and with shorter pulses). I would also re-check the conversion for Refs. 103 and 108, which report luminances of $1.5\text{-}2,000,000\text{ cd}/\text{m}^2$, which taking peak spectral values gives me intensities of approximately $10\text{W}/\text{cm}^2$.

Minor comments:

-The role of the MoO_3 buffer layer is unclear.

-Why was 2.0 chosen as the out-coupling simulation cut-off (Methods)? Evanescent waves beyond this wavevector might still couple to the electrodes and thus lead to dissipated power, but not to emitted power (I do not think this will be the case, but the choice should be justified-the cutoff is effectively fixed by the SPP mode frequencies and the physical separation for lossy surface waves).

-Why is there a wiring electrode for anode 1 *and* 2 in Fig. 1c?

-In Fig 3c and d x-axis titles should be electrically and optically pumped. The x-axis title on the right panels is difficult to see.

-I recommend tidying up the text for readability and clarity.

In conclusion, I think that this work is technically sound (barring some fixable issues with the manuscript). While the work does not present any major advances in high-brightness OLEDs or organic laser design, it does make major advances in integration, which allow the authors to realize a remarkable device allowing for OLED-driven lasing of an optically-pumped organic semiconductor laser. This is such an important result that I do think it will be of interest to the readership of Nature.

Referee #2 (Remarks to the Author):

Organic semiconductor laser (OSL) and OLED architectures have been extensively studied for more than 30 years since the first reports of F. Hide et al. in 1997 and C.W. Tang et al. in 1987, respectively. This paper is an exciting report on the extremely well-elaborated organic laser device architecture demonstrating lasing by indirect excitation of an organic lasing media (OSL) by an OLED. The original device architecture having multiple layers was accomplished after overcoming many technical issues such as coupling the light into the lasing area and decreasing the device resistivity including wiring, in addition to the very short pulse driving method of 5 ns. All technological integration is impressive, realizing indirect pumping of the PPV polymer-based OSL by the fluorene-based OLED. Technologically this is a significant breakthrough in organic optoelectronics. Further, the estimated current threshold is reasonably consistent with the previously reported OSLED by Sandanayaka et al. (Appl. Phys. Express, 2019).

1) The title should be changed. The title of "An electrically driven OSL" misleads to the direct pumping of an organic active layer by current injection. This device uses an indirect pumping method. I think it is important to separate direct and indirect pumping clearly in the title.

2) The fundamental I-V-E QE characteristics are missing. The DC I-V-EQE and pulsed I-V-EQE should be provided. I think the breakdown behavior in both DC and pulse operation should be provided. In particular, I expect the unique rolloff behavior with short pulse operation. Maybe the comparison of different pulse duration provides a unique exciton deactivation mechanism from the aspect of joule heating, electrical field quenching, or triplets contribution.

3) I think two important papers would be cited fairly in the document.

CW Tang's first OLED paper, APL (1987)

F. Hide's first polymer laser, Science (1997)

4) The refractive index values of each layer should be summarized, indicating the confinement of light in the OSL layer.

Referee #3 (Remarks to the Author):

Summary

Achieving electrical pumping of organic semiconductor lasers has long been considered a formidable challenge in organic electronics, since the first attempts more than 25 years ago. Here the authors use an approach based on an indirect electrical pumping, that is in fact an optical pumping by an Organic Light-Emitting Diode (OLED), which has never been shown before. To achieve this goal, the most important challenge is to fabricate an OLED that has a high enough exitance (power per unit

area) to reach the threshold of organic lasers, which should be between several tens to around a hundred of Watts per cm^2 according to the present state-of-the-art. In a previous paper this group had shown that inorganic LEDs could be used for organic laser pumping, but these sources can reach peak exitances in the kW/cm^2 range, whereas OLEDs, especially in the blue, are limited to a few Watts per cm^2 . In order to realize this breakthrough, the authors had to combine several elements : a) a very good laser material (BBEHP-PPV) ; b) an optimized so-called “sub-structured” distributed feedback resonator (which was the one used to demonstrate the lowest reported threshold to date of $15 \text{ W}/\text{cm}^2$ in an organic laser) ; c) an OLED that is directly connected to the organic laser part through high-index materials to avoid the outcoupling losses to air ; d) a clever approach to combine the OLED and the laser involving pressing through a parylene interlayer ; e) a blue OLED that is pushed to an exitance of $46 \text{ W}/\text{cm}^2$ thanks to a short pulse operation of only 5 ns, doped transport layers and a very efficient low-lifetime spirobifluorene material. The laser presents a threshold of $2,83 \text{ kA}/\text{cm}^2$, and interesting experiments are presented in the end of the paper that compare optical, electrical and mixed optical/electrical pumping of the same device.

General opinion

This study presents the first laser to be optically pumped by an all-organic incoherent source (that it itself electrically excited), but this does not make this device an electrically-pumped laser. Indeed, the title chosen for the paper is in my opinion misleading. In semiconductor laser physics, the term “electrical pumping” clearly refers to as a process in which the population inversion is created by the injection of a current into the device, which is not the case here. Although terms like “electrical injection” or “organic laser diode” are safely avoided, the term “electrically-driven” chosen in place of “electrically-pumped” contributes to make things unclear, as it suggests that the device is not indeed electrically pumped but rather just “driven” by electricity, which is obviously indeed the case of any laser at some point. I think this contribution would better be described as an indirect electrical pumping of an organic semiconductor laser by an organic light-emitting diode.

Even though the title is certainly not adequate, I think this work is very important as 1) it sets a record for the peak exitance of an OLED in the blue around $50 \text{ W}/\text{cm}^2$, reaching a level where OLEDs can be used as pump sources for lasers; 2) it paves the way towards an organic laser device that would be fully realized with organic electronics technologies. There is still obviously a lot of work ahead to make it a practicable device (here both evaporation and solution-process are used for the OLED and the laser respectively, and the final structure consists in two half-devices that are mechanically pressed together), but one can consider that the direction is now shown for exciting future research.

This work certainly deserves being published in a high-impact journal, however I am not sure that Nature is the most suited journal for this paper, considering the two following arguments:

1) This is not a demonstration of an electrically-pumped organic laser, as shown before.

2) Optical pumping by an OLED has been rendered possible here by an extremely clever combination of state-of-the-art concepts in chemical, electrical and photonic engineering, gathered all at once in a single device. This breakthrough has not however been made possible by any disruptive novel concept, as the molecular design, the short pulsed operation of OLEDs or the low-threshold organic laser had been published separately before.

These two reasons bring me to the conclusion that although this work will be of an indisputable

interest for the organic electronics community, it might have a more restricted impact for the rest of the scientific community. I would then recommend a publication in another journal such as Nature Communications or Nature Photonics.

Questions and remarks on specific points

The paper is excellent and is written in an exquisitely simple, precise style. I wish the paper could contain more information about the device, notably its laser beam characterization. The device fabrication requires many steps, so it would be useful for any group willing to reproduce the data to have some practical information such as : how many devices were needed for one device to work ? Were the devices reproducible ?

I have a few specific remarks or questions that should be addressed :

- Line 42 : The introduction nicely puts the work in perspective and reminds that up to now almost all organic lasers were pumped by other lasers. The authors should however mention here their own work, cited hereafter as ref 27, on the indirect pumping of organic lasers by inorganic LEDs. This would better show how this study is a contribution in the reduction of the power density needed to operate organic lasers.
- Line 96 : The choice of a double layer of parylene and nanolaminates is not justified and the reader is sent back to ref 29 : a brief description of the reason of this choice, which seems very complicated at first sight, would be welcome
- Line 121 : It is not clear why the high current density of 10 kA/cm^2 is rejected from the data of ref. 30 : is there a limit in current density that the device should not overcome, based for instance on degradation arguments ?
- Line 147 : There should be a comment on the differences between short optical pulses obtained from an LED and an OLED : why is it easier to obtain shorter rise times compared to LEDs ?
- Line 225 : as explained before, I don't agree with the term "electrical pumping" used here. Actually there is no reason that optical pumping by OPO or by OLED would be that different provided that they have similar durations and that the absorption is the same.
- Fig 1 : The $130 \mu\text{m}$ distance represented by the double arrow is not consistent with the direction shown for the DFB grating grooves, this is a little bit confusing
- Fig 2 : I think the characterization of the OLED could be more complete. In particular there should be an I-V curve obtained in CW conditions, and compared maybe with the same under pulsed operation.
- For fig 2d and extended data fig 3, it should be mentioned what are the papers, among the extended OLED literature, that were selected to make this literature survey. Was it limited to OLEDs emitting in pulsed mode (and if so, what was the limit of pulse duration ?), or was it limited to deep-blue OLEDs ?
- Fig 3 : although evidence of lasing is supplied without too many doubts, I find this laser characterization a little unsatisfactory.
 - o Figs. 3 c and d do not really prove lasing, they just show a beamlike pattern that could be hidden in noise below threshold because of a lower power. A clear proof would be to show the photonic band diagram of the surface-emitting DFB laser, which should exhibit a collapse of both angular and spectral spread above threshold.
 - o The polarization data in fig 3b should be compared with the fluorescence polarization recorded below threshold, as a diffractive structure like a DFB grating yields polarized luminescence branches

below threshold usually

o Is the solid line in fig 3b a guide to the eye ? If so, is there a reason for the curve above threshold to be incurved downwards ?

- Extended data fig. 2 : there is probably a legend missing, I don't get what the dashed lines stand for
- Extended data fig. 4 : unless I did not read everything very carefully, I did not find experimental details relative to this loss measurement coefficient.

Referee #4 (Remarks to the Author):

Re: An Electrically Driven Organic Semiconductor Laser
by Yoshida, et al.

This manuscript reports an electrically-pumped organic laser. Although we still don't know what an organic laser might be good for, this result is a remarkable achievement that many considered to be impossible. It represents perhaps the culmination of 20-30 years of work on organic EL.

This is not the first report of an organic electrically-pumped laser (Chihaya Adachi, et al. have a publication from 2019). But that device was problematic in many ways. There are arguments about whether it truly describes a laser, and the device was very unstable, lasting for only a handful of pulses.

The approach by Yoshida, et al. is different, and given the significant step forward in performance, I think it is a notable advance conceptually as well as practically. This device features a separate OLED pump and optical gain region. The separation minimizes losses in the optical gain structure.

Technologically, the device is one of the most sophisticated structures ever built from organic semiconductors. There are 17 layers in the cross section shown in Fig. 1b. It combines a sub-structured grating, solution processed gain medium, and most importantly, an OLED capable of operation at nearly 10kA/cm².

I found the threshold, beam, and joint EL-optical measurements to be compelling.

I have only a couple of suggestions:

1. In Fig. 2a, the purple curves are linked to the right-hand axis, but it looks like this is unnecessary given that the spectra are all normalized.
2. Given the sophistication of the device it is not surprising that significant modeling was performed (described on pages 18-20). For the field to move forward, it will definitely need predictive modelling of these structures.

In particular:

(i) It would be very interesting to see graphical results from the EM simulations including the predicted spatial distribution of output power from the OLED and the overlap with the lasing mode. That would provide a simulation of the confinement etc... and a sense of the coupling efficiency between the OLED and laser. There is discussion on pg 10 about the coupling between the gain region and the pump in the integrated device relative to separated structures. But some sense of the absolute efficiencies, losses etc... would be very useful. If possible, a spatial plot of the losses would also be valuable.

(ii) I'm surprised that the authors assumed a Lambertian pattern especially given the peaky output of the OLED, which is presumably due to the weak cavity formed by the anode and cathode. Why was it necessary to assume a particular emission pattern? One imagines that the emission pattern could be calculated.

(iii) It would be interesting to compare the modelling results (especially the predicted coupling between OLED & cavity) to the observed threshold.

3. Readers may benefit from some context. The concept of a separated pump and lasing cavity has a long history in lasing. Diode pumping is used in many conventional systems today (especially high-power lasers?). In organics, the concept dates at least to PRB 66, 035321 (2002), perhaps earlier.

4. Finally, we have struggled since the early days of optically-pumped organic lasers to identify the unique advantages of organic EL lasers. The immediate impact of this work is to demonstrate that they are possible. The closing speculation about a few potential, and rather obscure, applications struck me as a potential distraction from the main result.

Author Rebuttals to Initial Comments:

Nature manuscript 2023-02-03436

Electrically driven organic laser using integrated OLED pumping

Kou Yoshida, Junyi Gong, Alexander L. Kanibolotsky, Peter J. Skabara, Graham. A. Turnbull*, Ifor D. W. Samuel*

Response to Reviewers' comments:

We are very grateful to the reviewers for their thoughtful, detailed and constructive comments on our manuscript. We have addressed all the points raised as explained below. In our response the original comments from the reviewers are presented in *sky blue italic style*, our responses are in black and changes in the manuscript are highlighted in *red italic style*. The numbering of figures follows the section-based numbering based on the Reviewer number (e.g., Figure 2.2 refers to the second figure presented in the response to Reviewer #2).

Reviewer #1 (Remarks to the Author):

1-1 The title is succinct, but I do not feel it correctly reflects the content of the work. One could also have used the same title for previous work from the same group using GaN LED or laser diode pumping. I suggest highlighting the fact that the structure is all organic, while also avoiding the insinuation that this is an organic laser diode.

Following the Reviewer's suggestion, we modified our title to be : "*Electrically driven organic laser using integrated OLED pumping*". We believe the new title clearly differentiates the paper from prior work, conveys it is all-organic and operated by electrical drive of the OLED.

1-2 The complete absence of uncertainties (especially given the topic) in the manuscript is somewhat shocking. Many values are given to 3 significant digits, with no uncertainty and not a single error bar can be found on the plots. Type A and B uncertainties on the various measured quantities should be reported/combined using statistical analysis and calibration reports and details of how uncertainties have been calculated should be reported.

Following the Reviewer's suggestion, we assessed type A and B uncertainties of the measured quantities in our experiments, and calculated the extended uncertainties with the coverage factor $k = 2$, which shows the interval of 95% confidence in the normal distribution. For example, we considered uncertainties in the calibration scale of the microscope for the OLED size measurements, the reading resolution of the oscilloscope, statistical variation in repeated measurements, as well as uncertainties in the linear fits of laser output below and above threshold, and their intersection, to estimate lasing threshold. Table 1.1 shows an example of uncertainties to measure current density, for which we previously expressed to 3 significant figures as the Reviewer mentioned. The extended uncertainties for the current density of 6.3 kA/cm^2 is $\pm 0.4 \text{ kA/cm}^2$ corresponding to a fractional uncertainty of 6%. This uncertainty is mainly due to the calibration scale in the microscope used for the OLED size measurement. We have now restated the current densities to 2 significant figures with uncertainties.

Table 1.1 Summary of uncertainty in current density measurement for the estimate of 6.3 kA/cm^2 .

Factors causing uncertainties	Probability distribution (Type of evaluation)	Standard uncertainty in the unit related to the factors (Fractional of uncertainty)	Uncertainty in the unit of the estimate (Fraction of uncertainty)
OLED size measurement	Normal (B) [1]	$1.477 \times 10^5 \mu\text{m}^2 \pm 0.045 \times 10^5 \mu\text{m}^2$ (3%)	0.19 kA/cm^2 (3%)
Oscilloscope vertical reading resolution	Rectangular (B)	$7.76 \times 10^{-2} \text{ V} \pm 0.044 \times 10^{-2} \text{ V}$ (0.57%)	0.035 kA/cm^2 (0.57%)
Repeating measurement	Rectangular (B)	$6.30 \text{ kA/cm}^2 \pm 0.57 \text{ kA/cm}^2$ (0.9%)	0.057 kA/cm^2 (0.9%)
Combined uncertainty			0.2 kA/cm^2 (3%)
Extended uncertainty (Coverage factor = 2)			0.4 kA/cm^2 (6%)

Note: [1] Combined uncertainty of OLED size measurement which includes uncertainties caused by minimum pixel size, minimum scale to calibrate length in the microscope image, and variation in the repeating measurements.

We have revised the values in the manuscript and its figures to show relevant uncertainties in our measurements. As an example, in Response Fig. 1.1 below we revised manuscript Fig.2c to show estimates of peak radiant exitance averaged over interpolated data of 11 pixels, and their extended uncertainties in both current density and radiant exitance. We note that in order to show variations of light output from different OLED pixels within the same fabrication run, we have changed our benchmark current density used for the maximum radiant exitance of the PNP-OLED to a lower value, i.e., from 6.7 kA/cm² (previous version of this paper) to 6.3 kA/cm²(this version).

We also added the following sentences in the label of Fig.2c.

Black symbols show the data of 4 different pixels fabricated in the same batch. Red symbols show estimates of peak radiant exitance of the PNP-OLED at 5 kA/cm² and 6.3 kA/cm², averaged over interpolated data of 11 pixels including the 4 pixels shown in the figure and 7 pixels from a different batch. Error bars are extended uncertainties of the measurement with a coverage factor of 2.

Also, we added a section in the Methods to describe evaluation of measurement uncertainties, ‘Measurement uncertainties in OLED and laser characterization’:

Uncertainties in our measurements are expressed as extended uncertainties for an interval of 95% confidence (coverage factor $k = 2$). These take account of combined uncertainties in the calibration of the energy meter and the scale of the microscope used for the OLED size measurements, the reading resolution and time resolution of the oscilloscope, pixel to pixel variations of OLED light output and variations in repeated measurements, as well as uncertainties in the linear fitting of laser output to estimate lasing threshold. The uncertainties in radiant exitance, EQE, and the laser threshold under optical pumping include a 10% calibration uncertainty of the energy meter. The uncertainty in threshold current density is mainly due to the calibration scale in the microscope used to measure the size of the OLED.

Response Figure 1.1, (Revised Fig. 2c). Black symbols shows the data of 4 different pixels fabricated in the same batch. Red symbols show estimates of peak radiant exitance of the PNP-OLED at 5 kA/cm² and 6.3 kA/cm², averaged over interpolated data of 11 pixels including the 4 pixels shown in the figure and 7 pixels from a different batch. Error bars are extended uncertainties of the measurement with coverage factor of 2.

1-3 The most basic measures of laser performance are the slope efficiency and wall-plug efficiency. There are no actual values on the y axis for any of the electrically-driven laser plots. This should be fixed.

We regard the most basic measures of laser performance to be threshold and slope efficiency, and agree these should be clearly stated in the paper. While the pulse energy should also be clearly stated, it does not need to be on all graphs, and we prefer to stay close to the quantity measured by the instrument. A similar approach has been taken in many high-quality laser papers in the Nature family that show power characteristics recorded with spectrographs in uncalibrated units (and many do not actually include calibrated efficiency):

- (1) Nature Photon **11**, 784–788 (2017). <https://doi.org/10.1038/s41566-017-0047-6> Perovskite-CW laser (Prof Giebink);
- (2) Nat Commun **11**, 271 (2020). <https://doi.org/10.1038/s41467-019-14014-3> Lasing from colloidal-quantum dot in LED like device structure (Dr Klimov);
- (3) Nature **585**, 53–57 (2020). <https://doi.org/10.1038/s41586-020-2621-1> Room temperature CW lasing from perovskites (Prof Adachi);
- (4) Nat. Photon. **14**, 452–458 (2020). <https://doi.org/10.1038/s41566-020-0631-z> Intracellular laser (Prof Gather);
- (5) Nat. Photon. **15**, 738–742 (2021). <https://doi.org/10.1038/s41566-021-00878-9> Lasing from quantum dot based on PbS (Dr Konstantatos)]

We have therefore added sentences on the pulse energy and slope efficiency of our laser in section 6 as follows:

We identify this change as the threshold current density of about 2.8 kA/cm². The FWHM of our laser for a current density of 4.9 kA/cm² is 0.09 nm, limited by the spectral resolution of the measurement system. The maximum output pulse energy was $(1.5 \pm 0.1) \times 10^{-5}$ nJ, and we calculate the slope efficiency (peak optical power / peak input current) of the laser to be 2.1 ± 0.2 μW/A. The laser efficiency is currently limited by two factors: a significant roll-off in OLED quantum efficiency under intense short-pulse operation and a low ratio of surface out-coupling to other losses in the DFB laser cavity. Further refinement of the cavity design, and a better understanding of the dynamics of OLEDs under nanosecond pulsed operation, should each lead to significant future improvements in laser efficiency.

We also added text in the ‘Electrically driven laser characterization’ section in Methods:

The output pulse energy and slope efficiency of the electrically driven laser were determined by calibrating the response of the CCD camera to a calibrated energy meter (J3S-10, Coherent); the organic laser pulse duration was measured to be 2.3 ± 0.1 ns using a silicon avalanche photodiode (APD430A2, Thorlabs).

We have not included wall-plug efficiency in the characterisation because the key result of this paper is to show we can exceed threshold. Optimisation of wall-plug efficiency will require developments listed above to increase slope efficiency and operate further above threshold.

1-4 I do not think it is appropriate for the authors to use 7 lines of their introduction to denigrate Ref. 26, in what seems like an attempt to increase the importance of their own work. I believe that simply highlighting that the diodes in the one report on OSLEDs are extremely short-lived (tens of pulses) and that yield was low is sufficient.

We were actually trying to give a balanced account of the work, and want to mention some of the positive features such as the clever materials design. We have reconsidered the 7 lines and shortened them to 5 lines as follows:

Adachi and co-workers found that the absorption spectrum of polarons and triplets of a carbazole based laser material did not overlap with the gain spectrum²⁴, so that one of the above problems could be overcome²⁵. They showed some features of lasing, including narrowing of the emission spectrum. However, the emitted beam was not very clear, and the yield of these devices was low (5%) and their stability was very poor (operated for 20 pulses above threshold).

1-5 It would be nice to include photostability data for the OLED to accompany Ext. Data Figure 8.

We have added operational lifetime data for the OLED to Ext. Data Fig.8.

1-6 The use of radiant exitance for the output intensity is somewhat esoteric. In the 1000s of papers on LEDs since the 1960s, I do not ever recall seeing its use for the EL Intensity (despite the radiometric meaning being correct). In the field of infrared LEDs, "emittance" is sometimes used, but is also less common than Intensity/Irradiance.

We agree that radiant exitance is not widely used, but it is the correct term for the key quantity in our study which is the power per unit area leaving the OLED. One reason for its rarity in the OLED field is that nearly all OLED work is for displays and lighting and so uses photometric units. Alternatives to radiant exitance such as intensity and emittance are used in several ways and so carry some ambiguity which we wish to avoid. Irradiance has the same units but is a different quantity – the power per unit area falling on a surface. In our integrated device, the distinction between power per unit area leaving the OLED and power per unit area

falling on the laser is very important and so requires a greater precision of terminology than may have been used in the past.

1-7 While the peak brightness (48W/cm² at 7kA/cm²) achieved from the OLED is quite good, the performance seems on par with other reports, which show peak brightness's of around 10W/cm² at 0.5-1kA/cm². The only difference is that the OLEDs in this manuscript were driven a little bit harder (and with shorter pulses). I would also re-check the conversion for Refs. 103 and 108, which report luminances of 1.5-2,000,000 cd/m², which taking peak spectral values gives me intensities of approximately 10W/cm².

We note that reference 103 (Ahmad, V. *et al.*) is now reference 100 in the revised manuscript; and reference 108 (Shukla, A. *et al.*) is now reference 105 in the revised manuscript.

We believe the calculated light output of the OLEDs in these publications are not around 10 W/cm². Although the detail of the calculation is described in the 'Literature survey of device performance and details of data collection'. Here, we show our calculation of their radiant exitances as an example step-by-step.

In ref. 103 (Ahmad, V. *et al.*), they achieved maximum brightness (L) of 3,000,000 cd/m². The peak wavelength of 550 nm is given in a different paper (Burns, S. *et al. Scientific Reports* **7**, 40805 (2017)). By assuming monochromatic light at 550 nm, we converted photometric units to radiometric units. We note that the photopic response at 550 nm is 0.9949. Finally, radiant exitance was calculated by expanding brightness to all emitting angle by assuming a Lambertian emission pattern, i.e., πL .

$$\text{Radiant exitance} = \frac{\pi \times L}{683 \times (\text{Photopic response at 550 nm})} = 13,900 \text{ W/m}^2 = 1.39 \text{ W/cm}^2$$

In ref. 108 (Shukla, A. *et al.*), they achieved L of 1,500,000 cd/m² and EQE (η_{EQE}) of 0.85% at a current density (J) of 90 A/cm². A peak wavelength (λ) of 565 nm is given in the same paper. By assuming monochromatic light at 565 nm, we converted the EQE and the current density to the radiant exitance. We note that we used EQE and current density instead of luminance to estimate radiant exitance as the values calculated from luminance may vary significantly depending on the emitter spectrum as mentioned in the Methods. The radiant exitance is calculated from production rate of photons estimated from EQE and current density and then considering the energy of each photon.

$$\text{Radiant exitance} = \frac{J}{e} \eta_{\text{EQE}} \times \frac{hc}{\lambda} = 1.68 \text{ W/cm}^2,$$

where h is the Planck constant, c is the speed of light.

We also note that it is not trivial to make OLEDs fast enough to respond to shorter pulses, and that our device has both double the radiant exitance of the previous record OLED (at any wavelength) and gives more than ten times higher light output than the previous record deep blue OLED in the region of 430 nm.

Minor comments:

1-8 The role of the MoO₃ buffer layer is unclear.

The role of this layer is to prevent damage of the active area when coated with epoxy. We mention this in 'PNPN-OLED fabrication' section in Methods.

1-9 Why was 2.0 chosen as the out-coupling simulation cut-off (Methods)? Evanescent waves beyond this wavevector might still couple to the electrodes and thus lead to dissipated power, but not to emitted power (I do not think this will be the case, but the choice should be justified-the cutoff is effectively fixed by the SPP mode frequencies and the physical separation for lossy surface waves).

We are confident that a normalised wavevector of 2.0 (twice the in-plane wavevector in the light-emitting layer of the OLED) is large enough to capture the evanescent waves in the device. This is illustrated in Response Figure 1.2 which shows our calculation of the dissipation spectra at 430 nm in the PNP-OLED. The evanescent SPP mode and lossy surface waves have negligible power beyond an in-plane wavevector of 1.2. In selecting the value of 2.0 for our calculations we followed the approach in Ref [52], but expanded the normalized wavevector range beyond their maximum value of 1.6.

Response Figure 1.2, Calculated dissipation spectra at 430 nm as a function of the normalized in-plane wavevector. The regions separated by the dashed lines indicate the regions of the device into which the emission is trapped (and dissipated).

Following the reviewer’s suggestion, we added a justification of cut-off in the ‘Calculation of out-coupling efficiency’ in Methods:

The simulation was conducted within a wavelength range from 300 nm to 800 nm with a 1 nm step and, at each wavelength, dissipation powers were calculated for normalized in-plane wavevectors from 0.0 to 2.0 with a step size of 0.002. This maximum in-plane wavevector was chosen to ensure that the calculation captures the dissipation into all evanescent modes in the device and is a wider range than previously used in related OLED calculations in Ref. 52. We confirmed that the evanescent SPP mode of the device has negligible power beyond this range. In the model, the PNP-substrate was included in OLED stack; optical constants of parylene and nanolamination layers were obtained from ref²⁹.

*1-10 Why is there a wiring electrode for anode 1 *and* 2 in Fig. 1c?*

We intended to refer to the different wiring electrodes that contact the anode (and cathode) rather than “anode 1 and 2”. For clarity, we have revised the label of the wiring electrode to “wiring electrode 1 for cathode”, “wiring electrode 2 for anode” and “wiring electrode 3 for anode”. We revised Fig.1 as shown in Response Figure 1.3 and the sentences in ‘PNP-OLED fabrication’ for clarity as follows:

On the PNP-substrate, 15-nm-thick molybdenum trioxide (MoO₃, Merck) was evaporated through a shadow mask. Then, 1.1 μm-thick aluminium(Al) was evaporated through another shadow mask (wiring electrode mask) to form wiring electrode 1 for the cathode and wiring electrode 2 for the anode (see Fig.1c). Aluminium was evaporated with a box heater with a crucible (EVCH5 and EVC5INTSPL01, Kurt J. Lesker) at rates of around 0.1 nm/s for the initial 100 nm and the rate was then gradually increased up to 2 nm/s.

After the HTL evaporation, the evaporation chamber was vented to swap and introduce materials for the following evaporation. Around 5 μm thick aluminium was evaporated to form the cathode and wiring electrode 3 for the anode, with another shadow mask (top electrode mask). The Al evaporation was split to three sections to prevent the OLED warming up during the evaporation.

1-11 In Fig 3c and d x-axis titles should be electrically and optically pumped. The x-axis title on the right panels is difficult to see.

We revised the axis labels to be 'electrically driven' and 'optically pumped' and made the x-axis title on the right panels larger as shown Response Figure 1.4.

1-12 I recommend tidying up the text for readability and clarity.

We hope that the revisions made to the manuscript improve its readability and clarity.

In conclusion, I think that this work is technically sound (barring some fixable issues with the manuscript). While the work does not present any major advances in high-brightness OLEDs or organic laser design, it does make major advances in integration, which allow the authors to realize a remarkable device allowing for OLED-driven lasing of an optically-pumped organic semiconductor laser. This is such an important result that I do think it will be of interest to the readership of Nature.

Reviewer #2 (Remarks to the Author):

2-1 The title should be changed. The title of “An electrically driven OSL” misleads to the direct pumping of an organic active layer by current injection. This device uses an indirect pumping method. I think it is important to separate direct and indirect pumping clearly in the title.

Following the Reviewer’s suggestion, we modified our title to be: *Electrically driven organic laser using integrated OLED pumping*. We believe the new title precisely describes our work, conveys the indirect pumping approach used in the integrated laser, and leaves space for future publications on injection lasers.

2-2 The fundamental I-V-E QE characteristics are missing. The DC I-V-EQE and pulsed I-V-EQE should be provided. I think the breakdown behavior in both DC and pulse operation should be provided. In particular, I expect the unique rolloff behavior with short pulse operation. Maybe the comparison of different pulse duration provides a unique exciton deactivation mechanism from the aspect of joule heating, electrical field quenching, or triplets contribution.

We have added example DC and pulsed J-V-EQE OLED data shown in the Response Fig. 2.1 (DC measurements were made on a device on a glass substrate, and nanosecond pulsed data using the PNP substrate used in the integrated laser) into Extended Data Fig.3. These show a significant roll-off in EQE under nanosecond pulse operation. We do not have breakdown data of the devices under different time scale operation but note that this is not needed to demonstrate lasing in our devices.

2-3 I think two important papers would be cited fairly in the document.

CW Tang’s first OLED paper, APL (1987)

F. Hide’s first polymer laser, Science (1997)

We are happy to add these important papers to the first sentence of the introduction:

Organic semiconductors consist of conjugated molecules which can be simply deposited by evaporation or from solution to make a range of electronic and optoelectronic devices^{1-7,12-15}.

2-4 The refractive index values of each layer should be summarized, indicating the confinement of light in the OSL layer.

We have added a graph showing refractive index spectra of the OSL materials to Extended Data Fig.5, as shown in Response Figure 2.2.

Reviewer #3 (Remarks to the Author):

3-1 This study presents the first laser to be optically pumped by an all-organic incoherent source (that it itself electrically excited), but this does not make this device an electrically-pumped laser. Indeed, the title chosen for the paper is in my opinion misleading. In semiconductor laser physics, the term “electrical pumping” clearly refers to as a process in which the population inversion is created by the injection of a current into the device, which is not the case here. Although terms like “electrical injection” or “organic laser diode” are safely avoided, the term “electrically-driven” chosen in place of “electrically-pumped” contributes to make things unclear, as it suggests that the device is not indeed electrically pumped but rather just “driven” by electricity, which is obviously indeed the case of any laser at some point. I think this contribution would better be described as an indirect electrical pumping of an organic semiconductor laser by an organic light-emitting diode.

We have changed the title of the paper to “*Electrically driven organic laser using integrated OLED pumping*”. We believe the new title precisely describes our work, conveys the indirect pumping approach used in the integrated laser, and leaves space for future publications on injection lasers.

3-2 The paper is excellent and is written in an exquisitely simple, precise style. I wish the paper could contain more information about the device, notably its laser beam characterization. The device fabrication requires many steps, so it would be useful for any group willing to reproduce the data to have some practical information such as : how many devices were needed for one device to work? Were the devices reproducible?

Integration of the OLED and OSL is an important process for the laser devices. PNP-OLEDs and BBEHP-PPV lasers with similar performance were routinely made with moderate yield (around 60%). However, when integrated, some did not show the characteristics of laser emission. We observed lasing in 3 out of 14 integrated devices, giving a yield of 20%. Thus ~5 devices were needed for one device to work. We measured the spatial distribution of the beam for two of the three devices, and both showed beam-like output.

Thus, we find that the lasing characteristics of the electrically driven organic lasers are reproducible, although threshold current densities differed between devices. This is probably because we assemble these devices manually and the applied pressure is not consistently controlled. This may explain why some devices did not show laser action. By improving the assembly process, the reproducibility may be improved in future. In addition, the available radiant exitance was very close to the threshold of the OSL. Small variations in the OLED performance will also contribute to the variation in the threshold currents.

Following this suggestion, we have added information on the device yield in the ‘PNPN-OLED Fabrication’ in Methods as follows:

Widths of the OLEDs are slightly different depending on the samples, 120-150 μm , while the lengths are similar (1 mm), probably due to the contact of the top shadow mask and the substrate. Thus, sizes of all samples were measured and used to estimate performance. PNP-OLEDs with similar radiant exitance were made with a yield of around 60%.

Also, we added sentences about the yield in the ‘Laser sample Fabrication’ in Method as follows:

Finally, a parylene layer of 1.5 μm thickness was deposited by the parylene coater as the contact layer. Lasers with similar threshold were made with a yield of around 60%.

Also, we added sentences in the same section as follows:

The integration was conducted in a clean room to avoid inclusion of particles between the OLED and the organic laser, which was found important for the stable operation. Integration of the OLED and BBEHP-PPV

laser is an important step in the fabrication that can affect the laser threshold current density. We tested 14 devices in this study and 3 showed narrow emission (yield of around 20%). We tested the spatial output of 2 of the devices, which both showed a similar beam-like spatial distribution of the output light as shown in Fig.3. The low yield is probably due to the manual application of pressure in the integration, which was not consistently controlled. By improving the assembly procedure, the reproducibility may be improved.

I have a few specific remarks or questions that should be addressed :

3-3 Line 42 : The introduction nicely puts the work in perspective and reminds that up to now almost all organic lasers were pumped by other lasers. The authors should however mention here their own work, cited hereafter as ref 27, on the indirect pumping of organic lasers by inorganic LEDs. This would better show how this study is a contribution in the reduction of the power density needed to operate organic lasers.

We thank the reviewer for the suggestion. We now mention Ref. 27 (now Ref. 26 in the revised manuscript) in the last paragraph of the introduction.

The gain medium is then excited by electroluminescence from the charge injection region²⁶. This is conceptually similar to diode laser pumped solid state lasers, and to nitride LED pumping²⁶⁻²⁸, but here we achieve a fully integrated all-organic device. In this way we avoid the losses due to injected charges, greatly reduce losses due to triplets, and also reduce losses due to contacts.

3-4 Line 96 : The choice of a double layer of parylene and nanolaminates is not justified and the reader is sent back to ref 29 : a brief description of the reason of this choice, which seems very complicated at first sight, would be welcome

The double layer of parylene and nanolaminates is used to give an enhanced barrier to oxygen and water compared with a single layer, as shown in Ref. 29. Following this suggestion, we added a description in the main text as follows:

The OLED and its 'PNPN-substrate' were then removed from the glass carrier for transfer onto the organic laser waveguide. The two pairs of P and N layers were used to give a better barrier to oxygen and moisture than would be provided by a single pair (Ref. 30). The laser comprised a 230 nm thick BBEHP-PPV layer deposited on a distributed feedback grating; a 2.2 μm cladding layer of poly(vinyl-pyrrolidone) (PVPy) and a coupling layer of 1.5 μm -parylene were coated on the BBEHP-PPV to complete the laser section.

3-5 Line 121 : It is not clear why the high current density of 10 kA/cm² is rejected from the data of ref. 30 : is there a limit in current density that the device should not overcome, based for instance on degradation arguments ?

We have examined Ref. 30 (now Ref. 32) and find current densities up to 4.5×10^6 mA/cm², i.e., 4.5 kA/cm², so no data are rejected from this paper. Please note that our comment on the need for 10 kA/cm² to achieve 50 W/cm² is not referring to data in this reference, but is our deduction of what would be required based on an extrapolation of the data in this reference.

We have clarified how we refer to this paper on page 5 where text now reads:

In reported OLEDs, radiant exitance around 20 W/cm² has been achieved at 4.5 kA/cm² with an external quantum efficiency of 0.2%.³¹ For this efficiency, a very high current density, over 10 kA/cm², would be needed to give 50 W/cm² light output. To compound the difficulty of injecting such a high current density, we also need emission in the blue.

3-6 Line 147 : There should be a comment on the differences between short optical pulses obtained from an LED and an OLED : why is it easier to obtain shorter rise times compared to LEDs ?

The GaN LEDs used in our previous work had a larger emission area of $\sim 1 \text{ mm}^2$, and so increased RC time constant may have limited the rise time. We have added a comment on the area to the manuscript:

We note that the obtained rise time is much shorter than the 6 ns rise time of the larger area ($\sim 1 \text{ mm}^2$) GaN LEDs used in our previous work^{26,33,36}.

3-7 Line 225 : as explained before, I don't agree with the term "electrical pumping" used here. Actually there is no reason that optical pumping by OPO or by OLED would be that different provided that they have similar durations and that the absorption is the same.

We are using the term "electrically" in the text & Fig.3c to distinguish between electrical and optical driving of our integrated device. We have changed the title of the paper to "*Electrically driven organic laser using integrated OLED pumping*" to convey more clearly that our device is indirectly pumped by an integrated OLED.

We have also revised the term 'electrical pumping' to 'electrically driven' in the main text as well as in the caption of figure Fig. 3c.

We use the comparison of OPO pumping and OLED pumping to estimate the irradiance on the polymer laser gain medium and hence the enhancement in OLED pumping by using the carefully integrated structure compared with the irradiance that could be achieved from a separate OLED pump.

3-8 Fig 1 : The 130 μm distance represented by the double arrow is not consistent with the direction shown for the DFB grating grooves, this is a little bit confusing

We have revised Fig. 1b to make the grating grooves much smaller to avoid the confusion as shown in Response Figure 3.1.

3-9 Fig 2 : I think the characterization of the OLED could be more complete. In particular there should be an I-V curve obtained in CW conditions, and compared maybe with the same under pulsed operation.

We are pleased to supply further OLED characterization. We measured the DC characteristics of the same OLED structure on a glass substrate and show these data (squares) in Response Figure 3.2. Under CW operation, however, the OLED cannot achieve the very high current densities that are required to drive the laser due to Joule heating and thermal breakdown. The pulsed characterization of the same OLED on PNP substrates is also shown in Response Figure 3.2. The range of these measurements was limited by the pulse driver electronics only to high voltages/current densities, although these do cover the range of Fig.2.c relevant to driving the laser. We have added these data to Extended Data Fig.3.

3-10 For fig 2d and extended data fig 3, it should be mentioned what are the papers, among the extended OLED literature, that were selected to make this literature survey. Was it limited to OLEDs emitting in pulsed mode (and if so, what was the limit of pulse duration ?), or was it limited to deep-blue OLEDs ?

The survey covers vertical organic devices with two terminals, which achieve high current densities of more than 10 A/cm². This includes both OLEDs and unipolar devices. It is not limited to pulse mode or to blue OLEDs. For clarity we revised the 'Literature survey of device performance and details of data collection' as follows:

The data used to plot Fig.2d and Extended Data Fig.3 are summarized in supplementary information and references [26,31,47,55-110]. *We collated performance characteristics of two-terminal vertical organic devices which achieve a high current density of 10 A/cm² or more. We included information on OLEDs and unipolar devices. In some cases, peak wavelengths were not reported in the publication. In such cases, peak emission wavelengths were used from other publications reporting either EL spectra or PL spectra of the same emission layers.*

3-11 Fig 3 : although evidence of lasing is supplied without too many doubts, I find this laser characterization a little unsatisfactory.

o Figs. 3 c and d do not really prove lasing, they just show a beamlike pattern that could be hidden in noise below threshold because of a lower power. A clear proof would be to show the photonic band diagram of the surface-emitting DFB laser, which should exhibit a collapse of both angular and spectral spread above threshold.

We agree a collapse of both angular and spectral spread above threshold should be included. Indeed, the collapse of spectral spread was already included in Fig. 3a which shows a distinct photonic stopband in the surface diffracted emission when driven below threshold (2.0 kA/cm²) and a spectral narrowing (down to 0.09 nm) when driven above threshold (4.9 kA/cm²).

We have adjusted the vertical scale of the far field emission plot in Extended data Fig.6 to more clearly see the optical emission pattern below threshold. As shown in Response Figure 3.3, when the sample was driven below threshold (2.0 kA/cm²), the emission from the sample showed a divergent spatial profile,

superimposed with two brighter stripes that are characteristic of the fluorescence diffracted out of the waveguide mode of a surface-emitting DFB laser. With this detectable fluorescent background, it is clear no laser beam was observed at this peak current density. However, when driven at 4.9 kA/cm^2 (above threshold), a clear narrow divergence laser beam was observed. The evolution of far-field emission image and beam profile show clear evidence of the collapse of angular spread when driven above threshold. Note that the signal to noise of our measurement would readily show narrowed emission at 2 kA/cm^2 if a linear process such as fluorescence shaped by a grating or cavity were responsible for the pattern visible above laser threshold.

To visualise the photonic band structure, we have also measured the angle-resolved photoluminescence spectrum of the laser sample. Response Figure 3.4 shows the photonic band structure of the laser sample. A photonic stop band was observed at around 541 nm. This result is consistent with the measurement in Fig. 3a. The laser wavelength of our electrically driven laser (542 nm) was located at the longer wavelength side of this photonic band edge which is typical in surface-emitting organic DFB lasers.

Response Figure 3.4, Angle resolved photoluminescence spectrum of the DFB laser when pumped below threshold which shows the characteristic anti-crossing of the waveguide modes due to the photonic stopband that gives rise to band edge lasing when pumped above laser threshold. (we note that the additional weak band observed in this figure is attributed to scattered light from another nearby grating on the sample during the measurement)

3-12 The polarization data in fig 3b should be compared with the fluorescence polarization recorded below threshold, as a diffractive structure like a DFB grating yields polarized luminescence branches below threshold usually

We have also measured the polarization of the emission below threshold. The sample was pumped by OPO (450 nm, 20 Hz) below threshold ($0.8 P_{th}$). We observe that the fluorescence is partially polarized as the Reviewer mentioned. However, above threshold the polarisation ratio of the laser is much increased than the fluorescence as shown in the inset of Fig. 2b. This significant increase in polarisation ratio supports the evidence for laser action.

We have added the below-threshold data to Extended Data Fig.7

Response Figure 3.5 (revised Extended Data Fig.7), The normalized intensity as a function of the linear polarizer angle when the sample is electrically driven above threshold, and optically pumped 20% below threshold, the blue curve is a least squares fit to a $\sin^2(\theta)$ function; 90° is defined as the polarisation parallel to the grating groove direction.

3-13 Is the solid line in fig 3b a guide to the eye? If so, is there a reason for the curve above threshold to be incurved downwards?

The solid lines in Fig. 3b are linear fits to the data below and above threshold. We updated the figure caption to clarify this. The downward curve is due to the log scale of the axes, and is characteristic of the s-shaped curve of a laser power characteristic when plotted on a double-log graph. The plot on a linear scale is shown in Response Figure 3.6.

Response Figure 3.6, Integrated lasing intensity as a function of peak current density (linear scale). The blue lines are linear fits to the data below and above threshold.

3-14 Extended data fig. 2 : there is probably a legend missing, I don't get what the dashed lines stand for

We apologise for this error and have corrected Extended Data Fig. 2b as shown below in Response Fig. 3.7. The dashed line shows the calculated enhancement of the coupling efficiency into an outer medium of high refractive index, compared to emission into air. In the initial version the arrows were missing. The blue line indicates the situation for light coupling into the PVPy cladding of the laser.

3-15 Extended data fig. 4 : unless I did not read everything very carefully, I did not find experimental details relative to this loss measurement coefficient.

We have added the following text on the measurement of the loss coefficient in the 'Characterization of film samples' in the Methods:

To measure the waveguide loss, the pump beam was focused into a narrow stripe shape (2.3 mm by 130 μm) using a cylindrical lens. The end of the stripe was positioned near the edge of the waveguide. The pump stripe was moved away from the edge of the film and the emission from the edge was collected by a fibre-coupled CCD spectrometer. The emission intensity from the edge was fitted by $I = I_0 \exp(-\alpha x)$, where I_0 is a constant intensity, x is the distance of the stripe from the edge of the film and α is the waveguide loss coefficient.

Reviewer #4 (Remarks to the Author):

4-1 In Fig. 2a, the purple curves are linked to the right-hand axis, but it looks like this is unnecessary given that the spectra are all normalized.

Indeed, this was not necessary, so we have removed the arrows and revised Fig.2a as shown below.

4-2. Given the sophistication of the device it is not surprising that significant modeling was performed (described on pages 18-20). For the field to move forward, it will definitely need predictive modelling of these structures. In particular:

(i) It would be very interesting to see graphical results from the EM simulations including the predicted spatial distribution of output power from the OLED and the overlap with the lasing mode. That would provide a simulation of the confinement etc... and a sense of the coupling efficiency between the OLED and laser. There is discussion on pg 10 about the coupling between the gain region and the pump in the integrated device relative to separated structures. But some sense of the absolute efficiencies, losses etc... would be very useful. If possible, a spatial plot of the losses would also be valuable.

The laser gain medium is pumped by electroluminescence coupled into a large number of spatial and spectral substrate modes. It is therefore more appropriate to consider the emission coupled into the different layers in the structure based on the maximum wavevector within the light cone of each layer. Response Figure 4.2 shows the light output from the PNP-OLED at 430 nm as a function of the normalized in-plane wavevector. The regions separated by the dashed lines indicate the regions of the device into which the emission is

trapped (and dissipated). The losses in the structure (integrated over the full emission spectrum) are 8% in evanescent modes, 14% absorption in the semi-transparent electrode, 16% in waveguide modes and the parylene layer, leaving 62% within the light cone entering the PVPy cladding of the laser waveguide. We assume that in the integrated structure all this light entering the PVPy layer may be used to pump the polymer laser. For the case of a separate OLED and organic laser, however, only the light within the air light cone (in-plane wavevector < 0.5) can be utilised to pump the laser.

Response Figure 4.2, Calculated dissipation spectra at 430 nm as a function of the normalized in-plane wavevector. In the figure, the four regions separated by the dashed line indicate different optical modes, i.e., outcoupled light to PVPy layer, confined light by parylene in PVPy interface (parylene), waveguided mode (WG), and the evanescent light.

We have added Response Figure 4.2 into the Extended Data Fig.2, and the following text to the figure caption:

...In part b, stars show the values calculated for the PNPn-OLED with an additional 1.5 μm parylene and PVPy as the coupling layer and vertical lines show refractive indexes of the cladding layers at 430 nm. c, Calculated dissipation spectrum at 430 nm as a function of the normalized in-plane wavevector. The regions separated by the dashed lines indicate the regions of the device into which the emission is trapped (and dissipated). The losses in the structure (integrated over the full emission spectrum) are 8% in evanescent modes, 14% absorption in the semi-transparent electrode, 16% in waveguide modes and the parylene layer, leaving 62% within the light cone entering the PVPy cladding of the laser waveguide. We assume that in the integrated structure all of this light entering the PVPy layer may be used to pump the polymer laser. For the case of a separate OLED and organic laser, however, only the light within the air light cone (in-plane wavevector < 0.5) may be utilised to pump the laser.

4-3 I'm surprised that the authors assumed a Lambertian pattern especially given the peaky output of the OLED, which is presumably due to the weak cavity formed by the anode and cathode. Why was it necessary to assume a particular emission pattern? One imagines that the emission pattern could be calculated.

Following the Reviewer's suggestion, we calculated the irradiance falling on a surface illuminated by the PNPn-OLED using our out-coupling efficiency simulation. Response Figure 4.3 shows the new calculation, which is included as a revised version of Extended Data Fig. 1. The new calculation changes the peak irradiance at a distance 100 μm from the OLED from 55% of the peak radiant exitance (Lambertian profile)

to 60% of the peak radiant exitance. We have consequently updated the text mentioning the calculated results in the 'Overview of integrated OLED':

Emission from an OLED is normally highly divergent, so the irradiance decreases rapidly with distance. This is severe even over a short distance for OLEDs with a small active area; for the OLED of active area $130\ \mu\text{m} \times 1\ \text{mm}$ used in this work, at a distance of $100\ \mu\text{m}$ we calculate that the peak irradiance is reduced to only 60% of the peak radiant exitance of the OLED, while at $7\ \mu\text{m}$, the peak irradiance is higher than 99% (Extended Data Fig.1). We therefore designed the OLED and laser waveguide to be separated by only $7\ \mu\text{m}$ distance to maximise the excitation density in the gain material.

Furthermore, we added the following sentences to 'Calculation of irradiance distribution' in Methods:

In the simulation, the OLED was divided into $1\ \mu\text{m} \times 1\ \mu\text{m}$ sections and P_{OLED} was normalised to a value of 1. The OLED emission pattern was calculated by using the same simulation as the out-coupling efficiency. We simulated the emission pattern of the PNP-OLED emitting to parylene.

4-4 It would be interesting to compare the modelling results (especially the predicted coupling between OLED & cavity) to the observed threshold.

We have already presented a comparison of the modelled outcoupling efficiency (Ext. Data Fig. 2b) and the experimental results in section (6) *Characterization of integrated laser*. Using the assisted pumping measurement, we estimate the actual irradiance delivered to the organic laser from the OLED. Because the OLED and organic laser are very closely integrated, reduction in peak irradiance compared with the radiant exitance of the OLED is very small, less than 1%. Thus, the irradiance is expected to be same as the radiant exitance of the OLED emitted to the organic laser. By comparing this radiant exitance with the one measured for the OLED emitting to air, we estimated the enhancement of light coupling efficiency from emitting to air to emitting to the organic laser. We compared the enhancement with the calculated enhancement of out-coupling efficiency from emitting to air to emitting to the organic laser (Ext. Data Fig. 2b). To clarify our approach, we modified the text at the end of section 6:

The measurements of electroluminescence in Fig.2 show that for a peak drive current of 5 kA/cm², 43 W/cm² is outcoupled into air. The higher equivalent power density of 95 W/cm² in the integrated device suggests that the coupling efficiency to the gain medium is enhanced by a factor of 2.4 ± 0.3 (after taking account of differences in absorption between OPO and OLED excitation wavelengths), which is in good agreement with our calculated value of 2.3 for the coupling enhancement and demonstrates the benefit of our integrated device structure.

4-5 Readers may benefit from some context. The concept of a separated pump and lasing cavity has a long history in lasing. Diode pumping is used in many conventional systems today (especially high-power lasers?). In organics, the concept dates at least to PRB 66, 035321 (2002), perhaps earlier.

Following the Reviewer's suggestion, we revised the text in the third paragraph of the introduction:

In view of the difficulties outlined above, we have pursued an alternative approach in which we separate the region where charges are injected from the region where the laser population inversion is formed. The gain medium is then excited by electroluminescence from the charge injection region²⁶. This is conceptually similar to diode laser pumped solid state lasers, and to nitride LED pumping²⁶⁻²⁸, but here we achieve a fully integrated all-organic device. In this way we avoid the losses due to injected charges, greatly reduce losses due to triplets, and also reduce losses due to contacts.

4-6 Finally, we have struggled since the early days of optically-pumped organic lasers to identify the unique advantages of organic EL lasers. The immediate impact of this work is to demonstrate that they are possible. The closing speculation about a few potential, and rather obscure, applications struck me as a potential distraction from the main result.

We agree with the Reviewer's suggestion, and have removed speculation of future applications. Instead, we close by commenting on the implications for the field of organic optoelectronics:

This advance in organic lasers requires the OLED to operate under exceptionally intense current injection to make a new type of very fast organic optoelectronic device. The microscopic physics of OLEDs under such intense, short-pulse operation has been little explored to date. We anticipate that our work will stimulate future studies to understand the dynamics of organic semiconductors in this regime that could lead to significant improvements in device performance and open up new applications of ultrafast organic optoelectronics.

Reviewer Reports on the First Revision:

Referees' comments:

Referee #1 (Remarks to the Author):

I am satisfied with the response and find the revised manuscript much more rigorous than the previous iteration.

Referee #2 (Remarks to the Author):

I thank the authors for providing the appropriate answers except for one issue. I think the pulse duration dependence would be quite important for understanding the exciton deactivation processes. Please supply it with possible discussion.

Referee #3 (Remarks to the Author):

The authors have thoroughly answered all the questions raised by the reviewers, resulting in an appreciably better manuscript. I am happy to follow the opinion of other reviewers in considering this top-quality contribution as suitable for a publication in Nature.

However, regarding the main criticism made by almost all reviewers about the choice of the title, I still consider that the new title is not appropriate. The new proposition "Electrically driven organic laser using integrated OLED pumping" is misleading as the actual laser is not electrically pumped but rather optically pumped : it is of course "electrically driven", but exactly like basically all lasers are, except maybe some exotic sun-pumped lasers. Therefore, there is no reason to highlight the "electrical driving" of this laser, especially in the title, maybe except if the term goes with "indirect". As I already suggested in my first review, the term "electrical driving" is strongly associated with the idea of a population inversion obtained by electrical injection of carriers, and keeping the title as it is would maintain the ambiguity, which would not be totally honest. "An OLED pumped organic laser" or "Indirect electrical driving of an organic laser by integrated organic LED" or something with a similar flavor would be better titles, I think.

I have just another minor comment : My remark on fig1 was probably unclear ; I think the grating dimensions chosen in the first version were better for clarity than the smaller pitch chosen in the new version. My concern was rather about the dimensions of the device that appear in fig 1 : from Fig 1c, I understand that the DFB grating has dimensions 1 mm x 1 mm, but only a thin stripe of width 130 μm is active, the 130- μm dimension being parallel to the DFB grooves. But in fig. 1a, the 130- μm dimension appears to be perpendicular to the grooves. There is something confusing about fig 1a, in particular the significance of the purple trapezoid (is it representing light from the OLED ?)

Referee #4 (Remarks to the Author):

This is a remarkable technological achievement for organic semiconductors and OLEDs. The revised manuscript is exceptionally strong, with additions to the error analysis, lasing characterization, and modelling. I strongly recommend publication.

Author Rebuttals to First Revision:

Nature manuscript 2023-02-03436

Electrically driven organic laser using integrated OLED pumping

Kou Yoshida, Junyi Gong, Alexander L. Kanibolotsky, Peter J. Skabara, Graham. A. Turnbull*, Ifor D. W. Samuel*

Response to Reviewers' comments:

We are very grateful to the reviewers for considering our revisions based on their suggestions. We are also pleased to see that they are even more strongly supportive of the paper. We comment on the few and minor remaining issues below. In our response the original comments from the reviewers are presented in *sky blue italic style*, our responses are in black.

Reviewer #1 (Remarks to the Author):

I am satisfied with the response and find the revised manuscript much more rigorous than the previous iteration.

We are pleased to have been able to address the reviewer's comments.

Reviewer #2 (Remarks to the Author):

I thank the authors for providing the appropriate answers except for one issue. I think the pulse duration dependence would be quite important for understanding the exciton deactivation processes. Please supply it with possible discussion.

We have added as much data as we have for OLED characterisation in response to comment 2-2 in the first round of review. Unfortunately, our short current pulse driver (which generates the very high current density with fast rise time that is needed to access the driving regime for laser pumping) cannot systematically vary the pulse-width over an appreciable range, and so we cannot do the suggested study. We note that this level of characterisation of our OLED could be interesting, but is not needed to support our central claim of OLED-pumped lasing.

Reviewer #3 (Remarks to the Author):

The authors have thoroughly answered all the questions raised by the reviewers, resulting in an appreciably better manuscript. I am happy to follow the opinion of other reviewers in considering this top-quality contribution as suitable for a publication in Nature.

We are happy that the reviewer now considers the work to be of top quality.

However, regarding the main criticism made by almost all reviewers about the choice of the title, I still consider that the new title is not appropriate. The new proposition "Electrically driven organic laser using integrated OLED pumping" is misleading as the actual laser is not electrically pumped but rather optically pumped : it is of course "electrically driven", but exactly like basically all lasers are, except maybe some exotic sun-pumped lasers. Therefore, there is no reason to highlight the "electrical driving" of this laser, especially in the title, maybe except if the term goes with "indirect". As I already suggested in my first review, the term "electrical driving" is strongly associated with the idea of a population inversion obtained by electrical injection of carriers, and keeping the title as it is would maintain the ambiguity, which would not be totally honest. "An OLED pumped organic laser" or "Indirect electrical driving of an organic laser by integrated organic LED" or something with a similar flavor would be better titles, I think.

Based on the comments of the referees in the previous round of review we changed the title of the paper to "*Electrically driven organic laser using integrated OLED pumping*". There are several further points we would like to make:

1. We agree that other referees expressed concerns about the previous title and so we revised it. We note that the other three referees are now happy with the revised title, and so they consider we have resolved the previous potential ambiguity.
2. Our title clearly states that there is integrated OLED pumping, and so makes it clear that it is not an injection laser.
3. Our title leaves plenty of space for a future paper by another group to claim an injection laser. In fact we note that the paper by Prof Adachi 's team on "Indication of current-injection lasing from an organic semiconductor" explicitly used this term.
4. A crucial feature of our work is that the laser is integrated into the substrate of the OLED within a few μm of the electroluminescent layer. This creates a new type of integrated organic semiconductor device into which we input an electrical current and it outputs a laser beam. By analogy, white LEDs are considered to be electric lighting, although most of the photons emitted from a white LED originate from a yellow phosphor, which is integrated with a blue nitride LED into which the current is injected. Furthermore, a HeNe laser is widely considered to be an electrically driven laser even though the population inversion in the neon atoms is created via an intermediate energy transfer from excited-state helium atoms that were pumped by the electrical discharge.
5. We disagree that our device is "exactly like basically all lasers are". It is very different from past work using (for example) a microchip laser to pump a polymer laser which creates a much larger and bulkier laser system (approximately 10000 times larger volume). Other schemes commonly used to pump lasers such as flashlamps are much larger still. Even the microchip-pumped laser (or our previous work combining organic lasers with discrete nitride LEDs) could not readily be made into large arrays, cannot be flexible, and consists of multiple parts / optics needing alignment, so is distinctly different from our current approach.
6. We dislike the idea of attributing a very narrow meaning to "electrically driven" – we think the term should mean what it says in English. Actually, only imagining an injection laser as a result of this narrowness may have held back the field and stopped people thinking of alternative approaches such as ours.

For the above reasons we wish to keep the revised title of “*Electrically driven organic laser using integrated OLED pumping*”.

Reviewer #4 (Remarks to the Author):

This is a remarkable technological achievement for organic semiconductors and OLEDs. The revised manuscript is exceptionally strong, with additions to the error analysis, lasing characterization, and modelling. I strongly recommend publication.

We are very happy to see this referee’s enthusiasm for our paper.